# Gamma-Linolenic Acid (GLA) Protects against Ionizing Radiation-Induced Damage: An In Vitro and In Vivo Study

**DOI:** 10.3390/biom12060797

**Published:** 2022-06-07

**Authors:** Poorani Rengachar, Anant Narayan Bhatt, Sailaja Polavarapu, Senthil Veeramani, Anand Krishnan, Monika Sadananda, Undurti N. Das

**Affiliations:** 1BioScience Research Centre, Department of Medicine, GVP Medical College and Hospital, Visakhapatnam 530048, India; parepoorna88@gmail.com (P.R.); spolavarapu79@gmail.com (S.P.); 2Department of Radiation Biosciences, Institute of Nuclear Medicine and Allied Sciences, DRDO, Delhi 110054, India; anbhatt@yahoo.com; 3Quality Assurance Laboratory, Ship Building Centre, Vishakhapatnam 530014, India; vrsenthilmedphy@gmail.com; 4Department of Radiotherapy, Queen’s NRI Hospital, Vishakhapatnam 530013, India; anandkrishnan@gmail.com; 5Department of Biosciences, Mangalore University, Mangalore 574199, India; monikasadananda@gmail.com; 6UND Life Sciences, 2221 NW 5th St., Battle Ground, WA 98604, USA; 7Department of Biotechnology, Indian Institute of Technology, Sangareddy 502284, India; 8Department of Medicine, Sri Ramachandra Medical College and Research Institute, Chennai 600116, India

**Keywords:** radiation, prostaglandins, inflammation, lipoxin A4, cytokines, survival

## Abstract

Radiation is pro-inflammatory in nature in view of its ability to induce the generation of reactive oxygen species (ROS), cytokines, chemokines, and growth factors with associated inflammatory cells. Cells are efficient in repairing radiation-induced DNA damage; however, exactly how this happens is not clear. In the present study, GLA reduced DNA damage (as evidenced by micronuclei formation) and enhanced metabolic viability, which led to an increase in the number of surviving RAW 264.7 cells in vitro by reducing ROS generation, and restoring the activities of desaturases, COX-1, COX-2, and 5-LOX enzymes, TNF-α/TGF-β, NF-kB/IkB, and Bcl-2/Bax ratios, and iNOS, AIM-2, and caspases 1 and 3, to near normal. These in vitro beneficial actions were confirmed by in vivo studies, which revealed that the survival of female C57BL/6J mice exposed to lethal radiation (survival~20%) is significantly enhanced (to ~80%) by GLA treatment by restoring altered levels of duodenal HMGB1, IL-6, TNF-α, and IL-10 concentrations, as well as the expression of NF-kB, IkB, Bcl-2, Bax, delta-6-desaturase, COX-2, and 5-LOX genes, and pro- and anti-oxidant enzymes (SOD, catalase, glutathione), to near normal. These in vitro and in vivo studies suggest that GLA protects cells/tissues from lethal doses of radiation by producing appropriate changes in inflammation and its resolution in a timely fashion.

## 1. Introduction

The main aim of radiation therapy is to selectively kill cancer cells with minimal effect on the surrounding normal cells. Radiation not only kills or slows the growth of cancer cells, but it can also affect nearby healthy cells, causing significant side effects. Radiation is pro-inflammatory in nature, generating excess ROS, cytokines, chemokines, and growth factors with associated inflammatory infiltrates. In general, cells are extremely efficient at repairing radiation-induced DNA damage. It is well accepted that characterizing and predicting the effects of radiation on cells is challenging. Cellular response to radiation depends on the kinetics of different DNA repair processes, the spatial distribution of double-strand breaks, and the resulting probability and severity of misrepair. The survival of cells, or the type of cell death after exposure to radiation, depends on several factors, including cell type, radiation dose and quality, oxygen tension, *TP53* status, DNA repair capacity, cell cycle phase at time of radiation exposure, and the microenvironment [1]. While radiation therapy (RT) is a major cancer treatment modality, and is responsible for at least 40% of cancer cures, exposure to radiation may also damage normal cells and produce significant structural and/or functional loss. Protection of normal cells/tissues from radiation-induced damage is essential, not only to protect subjects from accidental radiation exposure, but also to preserve the surrounding normal cells/tissues. 

Ionizing radiation (IR) is an effective and commonly employed treatment in the management of many human malignancies. The ability of IR to control tumors relies on DNA damage; cellular DNA damage response and repair (DRR) processes are the key to determining tumor responses. DNA double-strand breaks (DSBs) generated by IR are a lethal form of damage, and are repaired via homologous recombination (HR) and/or nonhomologous end-joining (NHEJ) pathways. Efforts are being made to understand and exploit the differences in the repair pathways between tumors and normal cells in order to develop new methods of increased tumor cell killing that also reduce normal tissue injury and/or enhance their (normal cell) survival. 

Previous studies showed that radiation induces inflammation, enhances the generation of pro-inflammatory prostaglandins (PGs), and could reduce the synthesis of anti-inflammatory lipoxin A4 (LXA4), which may be responsible for the beneficial action of LXA4 against radiation-induced injury [2,3,4,5]. Radiation activates phospholipase A2 (PLA2), resulting in the release of arachidonic acid (AA), eicosapentaenoic acid (EPA), docosahexaenoic acid (DHA), and other unsaturated fatty acids from the cell membrane lipid pool [6,7,8], which are the precursors of a variety of pro- and anti-inflammatory molecules. Thus, AA is the precursor of pro-inflammatory PGE2, leukotrienes (LTs), and thromboxanes (TXs), and anti-inflammatory LXA4. PGI2 and PGJ2 [9,10,11,12,13]. This indicates that the ability of AA to regulate inflammation resides in the balance between the pro- and anti-inflammatory metabolites formed from it. LXA4 is the most potent anti-inflammatory compound that suppresses PGE2 and LTs formation, which have pro-inflammatory actions, and thus, induces the resolution of inflammation. Since AA is the precursor of several pro- and anti-inflammatory compounds, and local AA concentrations depend on the activity of PLA2, it is reasonable to suggest that factors that regulate its (PLA2) activity determine the degree and duration of the inflammatory process. This is evident from the fact that corticosteroids, which are suppressors of PLA2, are potent anti-inflammatory agents [14]. In this context, it is noteworthy that when the local concentrations of AA are low (i.e., in AA deficiency states), it leads to the formation of excess of PGE2 and LTs and the initiation and progression of inflammation, whereas optimal AA concentrations induce the formation of LXA4 [15,16], and consequently trigger anti-inflammatory events and the resolution of inflammation. In addition, the concentration of AA depends on the conversion of dietary LA (linoleic acid, an essential fatty acid) to AA by the action of desaturases [9,15,16]. Thus, the outcome of the inflammatory process, viz., its continuation or its resolution, depends on several factors, some of which include dietary LA, the activities of desaturases, PLA2, local concentrations of AA, the activity of COX-2 (cyclo-oxygnease-2 enzyme needed for the conversion of AA to PGE2 in response to inflammatory stimuli) and LOX (lipoxygenase enzymes needed for LXA4 formation) (see Figure 1). It is noteworthy that when the local concentration of PGE2 reaches optimum level, it triggers the initiation of the resolution of inflammation by blocking the formation of LTA4 and enhancing the activity of 5-LOX, such that the formation of LXA4 is increased (See Figure 1B). Thus, PGE2 may function both as a pro-inflammatory and anti-inflammatory molecule. This implies that the enhanced formation of PGE2 seen in inflammatory conditions and upon exposure to radiation, could be an attempt not only to trigger inflammatory events, but also to initiate its resolution once the inflammatory process reaches its peak [15,16,17,18]. This anti-inflammatory action of PGE2 is due to its binding to the prostaglandin E receptor 4 (EP4), resulting in the modulation of macrophage and T lymphocyte functions, which are crucial in innate and adaptive immunity and tissue remodeling and repair. The activation of EP4 suppresses the release of cytokines and chemokines from macrophages and T cells, inhibits the proliferation and the activation of T cells, and induces T cell apoptosis, events that ultimately lead to the induction of the resolution of inflammation [19]. This is supported by the report that 30 min prior, the administration of 16,16-Dimethyl prostaglandin E2 (DiPGE2), a stable analog of PGE2, increases the survival of mice against lethal doses of ionizing radiation. It is noteworthy that at 30 min after PGE2 injection, as much as 80% of the DiPGE2 was detected, unmetabolized, in the spleen and plasma, implying that the protection offered is due to the physiologic action of DiPGE2 [20,21]. Similar radioprotection was also reported with leukotriene C4 (LTC4) [22]. In this context, it is relevant to note that PGE2, and LTC4, LTD4, LTE4, and LTB4 (LTC4 > LTB4 > LTE4 > LTD4), derived from AA, are all considered as pro-inflammatory molecules, yet protect the gastrointestinal mucosa, liver, and pancreas from several injurious agents, and mouse intestinal stem cells from radiation injury [23,24]. Like LTA4, lipoxin B4 (which is similar in structure and function to LXA4) protected mouse hematopoietic stem cells against radiation [25]. 

Previous studies revealed that increasing PGE2 concentrations in the tissues by the inhibition of 15-PGDH (15-prostaglandin dehydrogenase, the PGE2-degrading enzyme) enhanced the regeneration of bone, colon, liver, and blood cells/tissues, and rejuvenated aged muscle mass and strength [26,27]. These results imply that PGE2 is needed for tissue regeneration, which may account for its radiation protection property. Thus, it is paradoxical to see that both pro- (PGE2 and LTs) and anti-inflammatory (LXA4 and LXB4) metabolites derived from AA, protected stem cells from radiation-induced damage in addition to enhancing the tissue regenerating ability of PGE2. In these studies [20,21,22,23,24,25,26,27], no efforts were made to study the role of LXA4. It is likely that enhanced concentrations of PGE2 and LTs can augment the formation of LXA4, as shown in Figure 1B, which, in turn, is able to mediate the beneficial actions of PGE2. Based on this interpretation, it is proposed that the formation of an adequate amount of PGE2 is needed to produce the optimum degree of inflammation; this could trigger the formation of LXA4, by redirecting AA metabolism, which is needed to initiate the resolution of inflammation, protect stem cells (which are needed for tissue regeneration), and restore homeostasis. If this proposal is true, it is necessary to study the dynamic changes in the concentrations of PGE2, LTs, and LXA4 in a suitable model of inflammation. Hence, in the present study, we evaluated the influence of various polyunsaturated fatty acids (PUFAs) on the survival of RAW 264.7 cells in vitro and C57BL/6J animals exposed to radiation, to understand how radiation alters PUFA metabolism and various eicosanoids and cytokines, and the expression of genes associated with inflammation, apoptosis, and metabolism, especially in the duodenum of animals exposed to lethal radiation. Thus, in this study, we evaluated changes in the activities of desaturases (needed for the formation of GLA, AA, EPA, and DHA from their precursors LA and ALA) and COX and LOX enzymes (needed for the formation of various eicosanoids, including LXA4), the concentrations of cytokines (including TNF, IL-6, IL-10, HMGB1) and anti-oxidants, and the expression of NF-kB, IkB, Bcl-2, Bax, iNOS, AIM-2, and caspase genes, and tried to correlate these changes to the survival of animals exposed to lethal radiation.

## 2. Material and Methods: Reagents

All culture media and additives were purchased from Sigma Aldrich Chemicals Pvt. Ltd., Bangalore, India. PUFAs and their metabolites were purchased from Cayman Chemical Company, Ann Arbor, MI, USA. It may be noted here that all the PUFAs used in the present study are given in μg or ng or pg and for the feasibility of their conversion to M, the molecular weights of all the fatty acids and their metabolites are given in Appendix A at the end of the text. 

### 2.1. Cell Culture Conditions

Murine monocyte/macrophage cell line (RAW 264.7), obtained from National Centre for Cell Science, Pune, India, was cultured in DMEM (pH 7.4) supplemented with bicarbonate, 100 U/mL penicillin, 100 µg/mL streptomycin, and heat-inactivated 10% fetal bovine serum (FBS) at 37 °C with 5% CO_2_. RAW cells were used in the undifferentiated state, grown as a monolayer culture, and sub-cultured upon reaching 80% confluency. Harvesting of the cells was performed by physical dissociation of the cells using a sterile cell scrapper, and a homogenous suspension was prepared by continuous pipetting. The cells were counted in a hemocytometer. 

### 2.2. Experimental Animals

The study was performed using 8–10-week-old C57Bl/6J female mice procured from the National Institute of Nutrition (Hyderabad, India). The animals were maintained at 25 °C with a 12 h dark and 12 h light cycle. Animals weighing around 15–20 g were segregated into four groups: 1. Control group, treated with 0.02% ethanol in PBS; 2. PUFAs group, receiving only GLA 100 μg/kg body weight intraperitoneally; 3. Irradiation (IR) group, treated with 0.02% ethanol in PBS, receiving 7.5 Gy; and 4. PUFAs + IR group, wherein animals received PUFAs (100 μg/kg) 48 h, 24 h, and 1 h prior to irradiation. Animals were maintained with access to standard mice pellet feed (Nutrimix, Nutrivet Life Sciences, Pune) and water ad libitum. The study was approved by the Institutional Animal Ethical Committee. 

### 2.3. Irradiation Procedure

Cells (in culture dishes) and animals were irradiated with CLINAC IX (Varian Medical Systems India Pvt. Ltd., Kalpataru Square, India, Unit 33, 3rd Floor, Off Andheri Kurla Road, Andheri East, Mumbai, Maharashtra 400059, India) at a dose rate of 1 Gy/min. Briefly, the animals were placed in a 35 cm × 35 cm (SSD- 100 cm) field in cages restrained with fiber mesh (tissue-equivalent material to avoid radiation scatter) to avoid climbing while allowing free horizontal movement inside the cage to minimize any stress. Animals were not anesthetized during the radiation procedure. The whole procedure was carried out under the supervision of a Radiation Safety Officer.

### 2.4. In Vitro Studies

#### 2.4.1. Exposure of RAW 264.7 Cells to Radiation

After overnight attachment of 0.5 × 10^4^ cells/200 μL of RAW cells plated in 96-well plates were exposed to different doses of radiation (1, 2, and 4 Gy). Cell viability was measured using the trypan blue method at 24 h and 48 h post-irradiation [28]. Cell survival was expressed as a percentage compared with the control.

#### 2.4.2. Study of Effect of PUFAs on Viability of RAW 264.7 Cells In Vitro

In total, 0.5 × 10^4^ cells/200 μL of RAW cells seeded into 96-well plates (after overnight attachment) were treated with different doses (250 ng/mL to 10 μg/mL) of various PUFAs (GLA, AA, EPA, and DHA) for 24 h, and their metabolic viability was evaluated using an MTT (3-(4,5-dimethylthiazole-2-yl)-2,5-diphenyltetrazolium bromide) assay, as described previously [29]. Metabolic viability was expressed as a percentage compared with the control in the same treatment group.

#### 2.4.3. Determination of Radio-Protective Efficacy of PUFAs by MTT Assay

In total, 0.5 × 10^4^ cells/200 μL of RAW cells seeded into 96-well plates were first left for overnight attachment, and then were treated with different doses of PUFAs (250 ng/mL, 500 ng/mL, and 1 μg/mL) for 24 h, following which they were exposed to radiation (2 Gy)—this is termed a pre-treatment schedule. These cells were assessed for their metabolic viability at 24 and 48 h after radiation using the MTT assay [29]. Metabolic viability was expressed as a percentage of the control employed in the same treatment group.

#### 2.4.4. Radio-Protective Efficacy of PUFAs by Growth Kinetics Assay

RAW cells (0.1 × 10^6^ cells/mL), plated in 35 mm culture dishes in triplicate, were left overnight for attachment. Subsequently, the cells were treated with various PUFAs (250 ng/mL) for 24 h, following which they were exposed to radiation (2 Gy). These treated cells were counted in a hemocytometer for viability using the trypan blue dye exclusion method [28] at 24 and 48 h. 

#### 2.4.5. Effect of PUFAs on Generation of Intracellular ROS 

Cells (0.1 × 10^6^ cells/mL) plated onto 35 mm culture dishes were left overnight for attachment. These cells were treated with various PUFAs (250 ng/mL) for 24 h, following which they were exposed to radiation (2 Gy). Subsequently, cells were washed with PBS and incubated for 30 min with dichloro-dihydro-fluorescein diacetate (DCFH-DA, 2 μM) at various time points (1, 4, and 24 h) post-irradiation. These PUFAs and radiation-treated cells were re-suspended in phosphate-buffered saline (PBS). Their intracellular ROS levels were measured at 488 nm using flow cytometry (BD LSRII, BD Biosciences) [30]. ROS levels were expressed as a percentage compared with the control. 

#### 2.4.6. Effect of PUFAs on Antioxidant Enzymes, Lipid Peroxides, and Nitric Oxide (NO)

RAW cells were plated and pre-treated with GLA (250 ng/mL) and exposed to radiation, as mentioned above. From these treated cells, the supernatant- and PBS (pH 7.4)-washed cells were collected at various time points (24 and 48 h) post-irradiation. The washed and collected cells were lysed with lysis buffer. Cell lysate was used for the estimation of the concentrations of antioxidant enzymes: catalase, superoxide dismutase (SOD), glutathione-S-transferase (GST), glutathione peroxidase (GPx), as described previously [31,32]. Concentrations of lipid peroxides ((such as malondialdehyde (MDA), formed upon reaction with thiobarbituric acid (TBA)) and nitric oxide (NO) (measured as nitrite formed using Griess reagent)) were estimated in both the supernatant and cell lysate, as described previously [31,32]. 

#### 2.4.7. Micronuclei Analysis

RAW 264.7 cells plated onto 35 mm culture dishes pre-treated with GLA (250 ng/mL) for 24 h and subjected to irradiation (2 Gy), as described above, were collected at various time points (4, 24, and 48 h) post-irradiation. These cells were fixed with Carnoy’s fixative (methanol and acetic acid in 3:1 proportion), and smears were prepared on glass slides and left for overnight drying, followed by staining with 4,6-diamidino-2-phenidole dihydrochloride (DAPI, 0.1 μg/mL) [33,34,35,36]. Cells were visualized under fluorescent microscope. The percentage of cells with micronuclei was calculated by counting 1000 cells per slide. 

#### 2.4.8. Gene Expression Analysis in RAW Cells

RAW cells, plated and pre-treated with GLA and radiation, as mentioned above, were collected at different time points (24 and 48 h after radiation) for RNA extraction using TRI reagent. cDNA was prepared by taking equal quantities of RNA (1 μg) using a RT-PCR kit (Superscript 1^st^ strand system for Reverse Transcriptase-PCR, Invitrogen). Semi-quantitative PCR was performed (Eppendorf 5331 master cycler) to study genes that are involved in GLA metabolism (COX-1, COX-2, D-6-dS, D-5-dS, and 5-LOX enzymes), genes involved in the inflammatory pathway (TNF-α, TGF-β, NF-κB, IκB, and iNOS), and genes responsible for apoptosis (p53, Bcl-2, Bax, AIM-2, caspase-1, and caspase-3), using primers (Bioserve, Hyderabad, India) (Table 1), as follows: Initial denaturation at 94 °C for 2 min, denaturation at 94 °C for 30 s, 30-s annealing, extension at 72 °C for 30 s, and final extension at 72 °C for 5 min. The overall cycle number was 35 cycles. The total reaction volume was 25 μL, with 1 μL cDNA, 2.5 μL 10X Taq buffer, 1.0 μL 10 mM dNTPs, 0.5 μL forward and reverse primers (100μM), 0.5 μL Taq DNA polymerase (3 U/μL), and 19.5 μL DEPC-treated water. PCR products were observed and analyzed by electrophoresis on 2% (*w*/*v*) agarose gel in 1X TAE buffer at 70 V for 30 min. Quantification (Major Science image analysis software) was performed by taking the ratio of gene of interest and GAPDH, and expressed in percentage compared with the respective control.

### 2.5. In Vivo Studies

#### 2.5.1. Dose Standardization of GLA for In Vivo Studies

Female C57BL/J6 mice were separated into eight groups containing 12 animals each for dose standardization studies with GLA (10, 50, and 100 μg/kg body weight). Vehicle (0.02% ethanol in PBS) and GLA were administered to the animals as intraperitoneal injections at 48, 24, and 1 h prior to radiation (7.5 Gy at 1 Gy/min). The best radio-protective dose of GLA was used for further studies. It should be mentioned here that our previous studies revealed that GLA increased survival only of female mice, with no significant change in the mortality of male mice compared to the radiation control. Hence, all further studies were performed with female mice.

#### 2.5.2. Survival Study for Radio-Protection with PUFAs and LXA4

Female C57BL/J6 mice were separated into 10 groups containing six animals each for survival studies with PUFAs (DGLA, AA, EPA, DHA at 100 μg/kg body weight). Vehicle (0.02% ethanol in PBS) and PUFAs were administered to the animals as intraperitoneal injections at 48, 24, and 1 h prior to radiation (7.5 Gy at 1 Gy/min). Survival and body weight were monitored for 30 days post-irradiation to evaluate the radio-protection offered by PUFAs.

Similarly female C57BL/J6 mice were separated into eight groups containing six animals each for survival studies of LXA4 pre-treatment at 10, 50, 100 ng/kg body weight, respectively. Vehicle (PBS) and LXA4 were administered to the animals as intraperitoneal injections at 48, 24, and 1 h prior to radiation (7.5 Gy at 1 Gy/min). Survival was monitored for 30 days post-irradiation to evaluate the radio-protection offered by LXA4 pre-treatment. In parallel, female C57BL/J6 mice were separated into four groups containing six animals each for survival studies of LXA4 post-treatment at 50 ng/kg body weight. Vehicle (PBS) and LXA4 were administered to the animals as intraperitoneal injections on day 7, 14, 21, and 28 post-radiation (7.5 Gy at 1 Gy/min). Survival was monitored for 30 days post-irradiation to evaluate the radio-protection offered by LXA4 post-treatment.

#### 2.5.3. Mechanism of GLA-Mediated Radio-Protection Studies 

To understand the mechanism(s) driving GLA response to radiation, female C57BL/6J mice were segregated into four groups, with 6 animals per group. These studies were performed at four different time points, viz., day 1, day 3, day 7, and day 14. In these studies, we used a dose of 100 μg/kg GLA to study its potential beneficial action against radiation-induced acute sickness. Mice were euthanized at the end of day 1, day 3, day 7, and day 14 post-radiation. Duodenum tissue was dissected to study various indices. 

Antioxidant status in mouse duodenum

Antioxidant enzymes: catalase, SOD, GPX, GST, and nitric oxides and lipid peroxides were measured in homogenized duodenum, as described previously [31,32].

b.Measurement of inflammatory and anti-inflammatory mediators in duodenum

To understand the role of pro- and anti-inflammatory cytokines as mediators of radiation-induced tissue damage and recovery of duodenum from radiation-induced injury, respectively, we measured plasma and duodenal concentrations of HMGB1 (201205), PGE2 (514010), and LTE4 (520411) by ELISA kits obtained from Cayman Chemical Company, Michigan, USA; lipoxin A4 (EA45) from Oxford Biomedical Research Company, MI, USA; IL-6 (431301), IL-10 (431411), and TNF-α (430901) from BioLegend, California, USA, as per the manufacturer’s instructions.

c.Gene expression studies on mouse duodenum

For this purpose, 50 mg of duodenum collected at various time points were treated with TRIzol reagent, and RNA was extracted. cDNA was synthesized and used to study the expression of Bcl-2, Bax, D-6-dS, COX-2, 5-LOX, Nf-κB, IKB, and GAPDH. Primer sequences are mentioned in Table 1. 

d.Protein expression studies on mouse duodenum

Protein expression analysis of duodenum was performed by Western blot after homogenizing the tissue in ice-cold lysis buffer containing 76.5 mM Tris, 10% *v/v* glycerol, 2% SDS, 200 mM sodium orthovanadate, and protease inhibitor cocktail. The lysate was incubated in ice for several minutes and centrifuged at 13,000 rpm for 5 min to remove debris. Subsequently, 40 μg of protein samples were loaded onto SDS-PAGE gel, and electrophoresis was performed at 100 V for 1 h. The separated protein samples were transferred onto PVDF membrane using a Bio-Rad blotting system (30 V at 4 °C) overnight. Blocking was carried out using 5% fat-free skimmed milk powder, and the membrane was incubated with primary antibodies (Bcl-2, Bax, Nf-κB, IκB, 5-LOX, COX-2, and GAPDH) at 1:1000 dilution for 12 h at 4 °C. Membranes were washed and incubated with secondary antibody (1:20,000 dilution) for 60 min. Membranes were developed after addition of Immobilon Western Chemiluminescent HRP Substrate for 1 min, and expression was captured using X-ray films under complete darkness. Further densitometry was performed using Major Science image analysis software. Protein expression was normalized using GAPDH as the housekeeping gene.

e.Histopathology of duodenum

Mouse duodenum was dissected 4 cm below the pyloric junction, washed in PBS, and fixed in 10% paraformaldehyde. Sections were embedded in paraffin, and 5 μM-thick sections were stained using hematoxylin and eosin (H&E) and examined under a microscope (Lawrence and Mayo) at 10× magnification. Height and width of the villi were measured and used to assess the acute GI damage induced due to irradiation and GLA-mediated protection.

### 2.6. Statistical Analysis

Data analysis was performed using MS Excel statistical analysis software. Each experiment was carried out in triplicate on two different occasions. The values are presented as mean ± SEM. *p* < 0.05 is considered as statistically significant.

## 3. Results

### 3.1. In Vitro Studies

#### 3.1.1. Effect of Radiation on RAW 264.7 Cells

Exposure of RAW 264.7 cells to different doses (1, 2, 4 Gy) of radiation caused a dose-dependent decrease in cell survival at both 24 and 48 h post-radiation (Figure 2A). Exposure to 2 Gy resulted in ~50% cell survival (LD_50_), which was used in all further studies.

#### 3.1.2. Effect of PUFAs on Viability of RAW 264.7 Cells 

PUFAs (GLA, AA, and EPA) were non-toxic at 250 ng/mL to 1 μg/mL concentrations to RAW 264.7 cells compared to control (Figure 2B). A dose-dependent decrease in the metabolic viability of cells at concentrations > 1 μg/mL, except for DHA, which was toxic even at 500 ng/mL, was noted. Hence in all future studies, 250 ng/mL to 1 μg/mL of various PUFAs were employed for their potential radio-protective action. 

#### 3.1.3. GLA Protects RAW 264.7 Cells from Radiation-Induced Toxicity

Pre-treatment with GLA and EPA (250 ng/mL) significantly (*p* < 0.001) improved the metabolic viability of irradiated RAW 264.7 cells at 24 h and 48 h compared to AA and DHA (Figure 3A–D). GLA at 250 ng/mL showed the highest protection (2-fold) at 48 h (Figure 3A). Radio-protective efficacy of PUFAs was further evaluated by analyzing radiation-induced growth inhibition by enumerating the viable cell count, which revealed a significant (*p* < 0.001) increase in the number of viable RAW 264.7 cells upon pre-treatment with GLA at both 24 and 48 h post-irradiation (Figure 3E).

#### 3.1.4. Effect of PUFAs on Intracellular Reactive Oxygen Species (ROS)

Radiation-induced increase in intracellular ROS levels in RAW 264.7 cells measured at 1, 4, and 24 h post-irradiation revealed that, of all the PUFAs tested, GLA induced the most significant (*p* < 0.001) decrease, both at 4 and 24 h (Figure 4). Hence, all further studies were performed using GLA. 

#### 3.1.5. Effect of GLA on Antioxidant Enzymes, Lipid Peroxides, and Nitric Oxide (NO)

Gamma radiation caused a significant (*p* < 0.01) decrease in SOD, catalase, GST, and GPx levels compared to control (C) at both 24 and 48 h post-irradiation. GLA pre-treated RAW cells (GLA + irradiation) showed improved antioxidant status and a significant (*p* < 0.01) increase in SOD, catalase, and GPx levels at both 24 and 48 h post-irradiation compared to radiation exposed cells. LPO/NO ratio was significantly high in irradiated cells compared to control (*p* < 0.01), at both 24 h (~1.5-fold) and 48 h (~4-fold) post-irradiation. The LPO/NO ratio in GLA-pre-treated cells was reverted to normal when compared to cells receiving irradiation at 24 h. A similar and significant decrease in the ratio of LPO/NO was noted (~2.6-fold) at 48 h post-irradiation (Table 2).

#### 3.1.6. GLA Decreased Radiation-Induced Cytogenetic Damage

The efficacy of GLA to prevent genetic damage in RAW 264.7 cells as assessed by the number of cells containing a micronucleus. This showed that radiation (IR) caused a significant (*p* < 0.01) increase in the percentage of cells with micronuclei at all post-radiation times (4, 24, and 48 h), while a significant (*p* < 0.01) reduction in the cells with micronuclei was observed in GLA-pre-treated cells. GLA alone did not increase the number of cells containing micronuclei (Figure 5A).

#### 3.1.7. PUFAs Modify Radiation-Induced Changes in Gene(s) Expression

Effects of GLA on radiation-induced alterations in the expression of genes associated with fatty acid metabolism (delta-6-desatuarse (D-6-dS), cyclo-oxygenase-1 (COX-1), delta-5-desaturase (D-5-dS), cyclo-oxygenase-2 (COX-2), and 5-lipoxygenase (5-LOX)); inflammatory pathways (tumor necrosis factor-α (TNF-α), transforming growth factor-β (TGF-β), nuclear factor kappa-light-chain-enhancer of activated B cells (NF-κB), NF-κB inhibitor (IκB), and inducible nitric oxide synthase (iNOS)); and apoptotic/anti-apoptotic pathways (p53, B cell lymphoma-2 (Bcl-2), Bcl-2-associated X protein (Bax), absent in melanoma-2 (AIM-2), caspase-1 (Cas-1), and caspase-3 (Cas-3)), were analyzed using semi-quantitative PCR. It is noteworthy that the expression of all these genes, which were altered by radiation, reverted to near normal levels in GLA-treated cells, as discussed below.

Metabolic pathway: COX-1 expression was significantly (*p* < 0.01) low in IR-treated cells compared to control (C) cells, which reverted to normal levels in GLA + IR-treated cells at 24 h. COX-2 levels, though significantly (*p* < 0.01) low in GLA + IR compared to IR cells at 24 h, was upregulated in GLA + IR compared to IR cells at 48 h post-radiation. 5-LOX levels were significantly (*p* < 0.01) low in GLA, IR, and GLA + IR cells when compared with control (C) cells at both 24 and 48 h post-irradiation. D-5-dS levels did not undergo any significant change in GLA + IR-treated cells compared with IR cells at both 24 and 48 h. D-6-dS levels in GLA + IR cells showed no significant change at 24 h, but were found to be subsequently enhanced (*p* < 0.01) compared to IR cells at 48 h post-irradiation (Figure 6). 

Inflammatory pathway: As expected, radiation significantly (*p* < 0.01) enhanced the ratio of TNF-α/TGF-β, which was reduced in GLA-treated cells at 24 h following radiation. In GLA + IR-treated cells, TNF-α/TGF-β ratio was significantly (*p* < 0.01) downregulated compared to IR (radiation alone) at 24 h, and subsequently increased at 48 h. IR significantly upregulated the NF-κB/IκB ratio (*p* < 0.01), while it was significantly (*p* < 0.01) lower in GLA + IR compared with IR treatment at both 24 and 48 h post-irradiation. GLA alone resulted in a decrease (*p* < 0.01) in NF-κB/IκB ratio at 48 h. There was no significant change in iNOS levels in GLA + IR-treated cells compared to IR cells at both 24 and 48 h post-irradiation (Figure 7). These results imply that radiation significantly upregulates genes associated with inflammatory pathways, which were dampened significantly by GLA treatment, suggesting that GLA can suppress inappropriate inflammation.

Apoptotic pathway: The Bcl-2/Bax ratio was significantly (*p* < 0.01) increased in GLA + IR compared with IR at both 24 and 48 h post-radiation, whereas IR induced a significant (*p* < 0.01) decrease in Bcl-2/Bax ratio compared with I control (C). In a similar fashion, radiation (IR) caused a significant (*p* < 0.01) increase in Cas-1 and Cas-3 expressions compared with I control (C) at both time points, which reverted to normal upon GLA pre-treatment (GLA + IR). p53 expression was significantly (*p* < 0.01) increased in IR compared to I control (C), which was significantly (*p* < 0.01) lower in GLA + IR-treated cells at 24 h; however, it (p53) reverted to normal at 48 h in both the groups. AIM-2 levels were significantly (*p* < 0.01) high in GLA + IR-treated cells compared with IR cells at 24 h, though its expression showed a significant decrease (*p* < 0.05) at 48 h (Figure 8).

### 3.2. In Vivo Studies

#### 3.2.1. GLA Protects Mice against Whole-Body Lethal Radiation

Our initial study revealed that of all the fatty acids tested, GLA was the most effective in protecting the animals from radiation-induced mortality (see Figure 9 for the protocol of the studies; the protocol is shown for GLA only, and the same protocol was used for studying the effect of other fatty acids, including LXA4. Figure 10 describes the control and GLA treatments (10, 50, and 100 μg/kg)). GLA and other fatty acids, including LXA4, employed in the study did not produce any change in the survival of C57BL/6J female animals during the period observed (see Figure 10). Irradiated mice showed 16.7% survival upon observation for 30 days. It is clear from the data shown in Figure 10 that GLA + IR-treated animals showed significantly increased survival compared to the IR group. Surprisingly, all other fatty acids tested (DGLA, AA, EPA, DHA, and LXA4) did not show any significant protective action. Of all the three doses of GLA tested, 100 μg/kg showed the best radio-protective effect, with 83.3% mice survival (Figure 10A) and significant increase in their body weight (Figure 10B). Hence, all further studies were performed using this dose. 

#### 3.2.2. Effect of GLA and Radiation on Cytokines in Mouse Duodenum

A sharp increase in HMGB1 levels was observed in IR mouse duodenal tissue on day 1, 3, and 7 compared to the control. GLA + IR treatment reverted HMGB1 levels to control levels by day 7, which showed further decline in concentration (*p* < 0.001) by day 14 post-irradiation (Figure 11A).

TNF-α concentrations in IR animals were significantly (*p* < 0.001) reduced by day 14 compared to the control; in contrast, TNF-α concentrations reverted to control values in GLA + IR animals (Figure 11B).

An insignificant increase in IL-6 levels was noted in GLA + IR mice compared to control mice on day1. IL-6 levels continued to decrease until day 14 in all the groups compared to the control (Figure 11C). 

No significant change in IL-10 levels was observed in IR mice except a significant increase on day 14 compared to control mice. GLA + IR animals exhibited significantly (*p* < 0.001) high levels of IL-10 on day 3 and 7 compared to IR mice, which reverted to near normal values by day 14 (Figure 11D). These changes in various cytokines suggest that there is a dynamic alteration in their concentrations that can vary from day to day. The most dramatic changes were found regarding TNF-α and IL-10, with both cytokines showing a significant increase in their concentrations on day 3. The concentrations of TNF-α reverted to near normal on days 7 and 14, with a sharp decrease on day 14 in the IR group, while IL-10 levels showed a dramatic increase. The GLA + IR group showed much lower concentrations of all cytokines studied, implying that GLA can suppress inappropriate changes in their levels, which could explain its beneficial action. 

#### 3.2.3. Changes in EFA Metabolism-Associated Genes in Mouse Duodenum

We next studied the effect of GLA and radiation on mRNA expression of genes associated with essential fatty acid (EFA) metabolism (D-6-dS, 5-LOX, and COX-2), inflammation (NF-κB and IκB), and apoptotic/anti-apoptotic pathways (Bcl-2 and Bax) in duodenal tissue of mice, employing semi-quantitative PCR. 

#### 3.2.4. Genes of EFA Metabolic Pathway 

COX-2 levels were significantly (*p* < 0.01) low in GLA + IR-treated animals compared to IR on day 1 and 3. Conversely, 5-LOX levels were significantly (*p* < 0.01) high in GLA-treated animals on day 1 and day 3. IR animals had significantly low levels of 5-LOX expression on day 3, and GLA + IR animals experienced an upregulation in 5-LOX levels on day 1 and 3, but these were reduced by day 14 compared with control levels. D-6-desaturase levels were significantly (*p* < 0.05) low in IR animals on day 1, 7, and 14 compared to control animals. GLA + IR animals exhibited significantly higher D-6-desaturase levels on day 3 and day 14 compared to control animals and irradiated animals, respectively (Figure 12). In view of the changes in the expression of COX-2 and 5-LOX enzymes, it is anticipated that significant changes may occur in the plasma and gut content/concentration of various eicosanoids. To verify this possibility, we measured gut (duodenal content) concentrations of PGE2, LTE4, and LXA4. 

#### 3.2.5. Effect of GLA on Radiation-Induced Changes in Eicosanoids

It is evident from the results shown in Figure 13 that there were dramatic and significant changes in the levels of PGE2, LTE4, and LXA4 in the duodenum. Radiation induced a significant increase in the concentrations of PGE2 (day 3 > day 1 ≥ day 7 ≥ day 14) associated with a concomitant decrease in levels of LTE4 on all these days. In contrast to this, LXA4 showed significantly elevated levels on all these days, with a peak on day 1 followed by a gradual and sustained decrease on days 3, 7, and 14 (day 1 > day 3 > day 7 > day 14) (see Figure 13C). It is evident from these results that in the GLA + IR group, PGE2 levels were close to control levels on all the days studied, while those of LTE4 were significantly higher on day 1, day 3, and day 7 (day 3 > day 1 ≥ day 7), and by day 14, they were much lower compared to the control. On the other hand, LXA4 levels were much lower compared to those seen in the radiation group on all days (Figure 13). The changes in the concentrations of PGE2, LTE4, and LXA4 in the GLA alone-treated group were similar to those seen in the GLA + IR group, except that they were much lower in magnitude. Thus, GLA treatment (both in GLA alone and GLA + IR groups) seems to induce changes in the levels of PGE2, LTE4, and LXA4 that are resemblant of those seen in the control group. Thus, GLA treatment seems to normalize altered eicosanoid levels. The alterations in PGE2, LTE4, and LXA4, especially in the GLA + IR group, seem to imply that a sustained elevation in LTE4, a muted change in PGE2 (almost no change except a slight increase on day 3), and a moderate increase in LXA4 levels, is needed to improve the survival of animals exposed to lethal radiation (see Figure 13). 

#### 3.2.6. Genes Associated with Inflammation

As expected, irradiation significantly upregulated the NF-κB/IκB ratio (*p* < 0.01) compared to the control. GLA + IR treatment significantly (*p* < 0.01) lowered the NF-kB/IkB ratio compared with the control and IR groups at all time points studied. (Figure 14). These results suggest that GLA can inhibit inflammatory process induced by radiation.

#### 3.2.7. Genes Associated with Apoptotic Pathways

Radiation is known to cause apoptosis/cell death. Hence, we studied its effect on the expression of genes associated with apoptosis and cell survival. Our results show that IR caused a significant (*p* < 0.01) decrease in the Bcl-2/Bax ratio on all days (1, 3, 7, and 14) of the study compared to the control. This indicates a significant degree of cell death in the duodenum tissues. In contrast to this, GLA + IR-treated mice showed a significantly (*p* < 0.01) higher Bcl-2/Bax ratio, indicating improved survival of cells against irradiation-induced insult at all time points (Figure 15).

#### 3.2.8. Protein Expression Studies on Mouse Duodenum

To confirm that the changes in gene expressions observed do indeed effect the formation of the concerned proteins, we studied concentrations of Bcl-2 and Bax by Western blot. Our results revealed that IR animals had significantly (*p* < 0.05) lower Bcl-2/Bax ratios compared to control animals on both day 1 and day 14. On the other hand, GLA + IR-treated animals exhibited significantly (*p* < 0.05) high Bcl-2/Bax ratios on day 1 compared to control and radiation-treated animals (see Figure 16). NF-κB/IκB levels were significantly (*p* < 0.05) increased on day 1 in IR and GLA + IR animals compared to control animals. On day 14, the GLA + IR group exhibited lower (*p* < 0.05) NF-κB/IκB ratios, indicating significantly decreased inflammation compared to the IR group. It is noteworthy that the expression of 5-LOX in GLA-treated animals increased on day 1 and 14, whereas COX-2 expression was significantly decreased on day 1, but showed remarkable enhancement on day 14 compared to control and GLA + IR groups (Figure 16). In comparison to this, 5-LOX expression was significantly decreased on day 1 and day 14, while COX-2 was significantly elevated on days 1 and 14, in the radiation group. All these radiation-induced changes in 5-LOX and COX-2 expressions reverted to near normal in the GLA + IR-treated group, except for the significantly elevated expression of 5-LOX on day 14 and the decreased expression of COX-2 on day 1, compared to the control and GLA + IR groups (Figure 16). 

#### 3.2.9. Histopathological Analysis of Mouse Duodenum

Since radiation-induced damage to the gut is one of the principal side effects that is responsible for significant morbidity and mortality, we next evaluated the histological changes in the duodenum of various groups. The duodenum of animals exposed to 7.5 Gy whole-body radiation showed significant shortening and rupturing of villi, lacteal shrinking, and mucosal denudation. The GLA alone treatment did not show any significant changes in the architecture and histopathology of duodenum morphology compared to the control. GLA pre-treated mice that were subsequently exposed to radiation showed significant changes in the duodenum, but a retention of morphological integrity, indicating that GLA promotes repair of intestinal tissue, and facilitates both the resolution of the inflammatory process and the healing of the damage induced by radiation treatment of the small intestine (Figure 17).

## 4. Discussion

Radiotherapy of cancer is common, and it often results in unintended injury to normal tissues surrounding the tumor, contributing to poor wound healing. Ionizing radiation causes DNA damage by direct strand breaks and free radicals. As a result, delayed wound healing occurs due to cellular depletion, stromal cell dysfunction, aberrant collagen deposition, and microvascular damage. Pro-inflammatory cytokines and free radical cascades are thought to be responsible for radiation damage, with transforming growth factor beta-1 (TGF-β1) acting as a key player in the process of fibrosis. The current radiation injury management has minimal success, and even surgical management is often compromised by the poor status of the surrounding tissue and recipient vasculature. Experimental treatment models currently employed include stem cell repletion, antioxidant therapy, TGF-β1 modulation, and implantable biomaterials, all of which show limited success. Hence, it is important to understand the complexity of wound healing and fibrosis following radiotherapy, and to develop potential solutions to this significant clinical and surgical problem [37]. In addition, exposure to lethal doses of radiation, especially during accidents in nuclear power plants, may lead to significant loss of life, severe radiation sickness, radioepidermitis, and other side effects. Internal radiotherapy (brachytherapy) uses high-dose intracavitary radiation or radioactive implants, and is used to deliver radiotherapy directly to prostate tumors, soft tissue sarcoma, breast tumors, and cervical tumors [38]. 

Adverse effects of radiation can be noted from the first days of radiotherapy, which depend on the radiosensitivity of the body sites being treated, the volume of normal tissue irradiated, the total dose and the rate of dose accumulation, with side effects most evident in rapidly proliferating tissues, such as the skin, mucosa, and bone marrow. Both acute (10–14 days after starting treatment) and long-term (more than 3 months after the end of treatment) effects of radiotherapy occur because of cell death within irradiated organs, and ischemia because of the irradiation of small blood vessels and/or perturbed inflammatory and repair responses. 

Acute inflammatory events seen after exposure to radiation are due to pro-inflammatory cytokines, such as tumor necrosis factor-α (TNFα), interleukin (IL)-1, IL-8, and interferon-γ (IFNγ) [39]. In general, termination of the inflammatory response is because of the short half-life of these cytokines, and production of anti-inflammatory cytokines, such as transforming growth factor-β (TGFβ), IL-4, IL-10, and IL-13. The inflammation triggered by radiation does not resolve adequately because of overproduction of pro-inflammatory cytokines, leading to perturbed cell–cell and cell–matrix interactions, uncontrolled matrix accumulation, and fibrosis [40]. In addition, nitric oxide (NO) synthesized in wounds by macrophages and fibroblasts promotes wound healing by inducing collagen deposition [41]. Previous studies suggest that NO is diminished in irradiated wounds, which may contribute to their poor healing capacity [42]. NO promotes collagen deposition, whereas TNF-α and IFN-γ prevent collagen deposition [43,44,45]; TNF and IFN are increased in irradiated wounds/tissues [42]. Thus, an imbalance in the secreted factors induces failure or poor wound healing following radiation, which may prove to be fatal at times [29]. Furthermore, irradiated tissues express decreased matrix metalloprotease 1 (MMP1), which causes poor wound healing by reducing cell migration, angiogenesis, and tissue remodeling [46].

Inflammation is the most predominant feature of radiation-induced injury. Hence, in the present study, we performed in vitro and in vivo experiments and measured various pro- and anti-inflammatory molecules (including cytokines) to determine the potential relationship between radiation-induced injury and inflammatory events, and how these molecules are altered during the inflammation resolution process, especially when lethal whole-body radiation is administered. It is noteworthy that a limited amount of work has been conducted on the role of bioactive lipids (especially polyunsaturated fatty acids and their pro- and anti-inflammatory metabolites) in relation to radiation-induced injury [2,3,4,6,7,8,20,21,22,23,24,25], which has led us to focus the current study on the affect(s) of various PUFAs on radiation-induced injury in vitro. It is evident from the results shown in Figure 2, Figure 3, Figure 4 and Figure 5 that of all the PUFAs tested, GLA is the most effective in protecting RAW 264.7 cells against radiation-induced DNA damage and apoptosis, at least, in part, by suppressing the generation of ROS. It is noteworthy that GLA treatment reduced the number of RAW 264.7 cells containing a micronucleus (or micronuclei) (see Figure 5), suggesting that this fatty acid (GLA) can reduce radiation-induced DNA damage and/or enhance DNA repair processes. These results are in support of our previous work, wherein we observed that GLA can protect RAW cells from radiation-induced cytotoxicity [47]. Our previous studies revealed that GLA, DGLA, and PGE1 protect mouse bone marrow cells and peripheral human lymphocytes from the cytotoxic action of benzo(a)pyrene, radiation, diphenylhydantoin (DPH), 4-α phorbol, and other chemicals [48,49,50,51,52,53,54,55], both in vitro and in vivo [47,48,49,50,51,52,53,54]. Similar results were obtained with studies using RIN (rat insulinoma) cells in vitro, and in streptozotocin- and alloxan-induced experimental diabetes models in rats [31,56,57,58,59,60]. Surprisingly, GLA, but not saturated or monounsaturated fatty acids (oleic acid), was more effective in preventing chemical- and radiation-induced damage to cells, whereas AA was more protective against alloxan- and streptozotocin-induced cytotoxicity to pancreatic β cells. This differential action of GLA and AA on different types of cells could be related to the tissue/cell-specific action of corresponding fatty acids. Thus, it is envisaged that each type of specific cell/tissue is endogenously protected by a specific fatty acid. This could be related to the way the fatty acids are metabolized by specific cells. In addition, in a previous study [55], we noted that oral supplementation of GLA to those who are on long-term diphenylhydantoin (DPH) therapy for epilepsy led to a decreased number of micronuclei-containing peripheral lymphocytes, but revealed a reduced DNA ladder pattern, an indication that DNA damage was increased. This apparently paradoxical action of GLA led us to suggest that GLA induces apoptosis of lymphocytes that harbor DNA damage. Based on these results, it was suggested that GLA functions as a unique endogenous molecule that protects cells from accumulating genetic damage by inducing apoptosis. This conclusion is supported by the observation that GLA can selectively induce the apoptosis of tumor but not normal cells by augmenting the accumulation of toxic lipid peroxides [61,62,63]. This argument is supported by the result shown in Table 2, wherein it can be seen that the balance between lipid peroxides and antioxidants is tilted more towards pro-oxidant status. Furthermore, both GLA and AA seem to possess the unique property of enhancing the proliferation of specific normal cells. In the case of pancreatic β cells, AA enhanced their proliferation by >10% even in the control [60], which is somewhat similar to the ability of GLA to augment RAW cell proliferation, as seen in Figure 3 of the present study (though this was not statistically significant). Thus, both GLA and AA possess the ability to enhance the proliferation of normal cells even under control conditions. This suggests that, in all probability, GLA and AA are needed for a normal proliferation index of cells/tissues. In contrast, both GLA and AA have been reported to suppress tumors, or even exhibit tumoricidal action [61,62,63,64,65,66,67,68,69,70], though GLA appears to be more specific and selective in its anti-cancer action compared to AA. 

To know how GLA influences the metabolism of EFAs (in the present instance, that of GLA), we next studied the changes in the activities of desaturases, COX-1, COX-2, and 5-LOX enzymes (the activity of 12- and 15-LOX was below the detectable limit) in RAW 264.7 cells. The results shown in Figure 6 clearly reveal that the activities of these enzymes were significantly enhanced in response to radiation and GLA + IR, compared to the control, at 24 h, whereas their activities reverted to near normal values at the end of 48 h, except for COX-2 activity, which was increased in GLA + IR-treated cells even at the end of 48 h. It is evident from these results that radiation produces acute changes and long-lasting actions on the activities of desaturases, COX and LOX enzymes, which is in line with its acute and chronic toxicity properties. These results can be interpreted to mean that the acute changes (at the end of 24 h) seen in desaturases and COX and LOX enzymes pertain to pro-inflammatory actions, whereas the substantial decrease (almost close to near normal levels) in their activities at the end of 48 h refers to the inflammation resolution process. This suggestion is supported by the results shown in Figure 7, which show that the ratios of TNF-α/TGF-β and NF-kB/IkB are significantly increased at the end of 24 h after exposure to radiation, but reverted to near normal levels at the end of 48 h. This indicates that acute radiation-induced inflammatory events (seen at the end of 24 h) tend to be resolved by 48 h. Furthermore, GLA + IR induced a substantial decrease in TNF-α/TGF-β and NF-kB/IkB ratios, both at the end of 24 and 48 h, suggesting potential anti-inflammatory actions of GLA and/or its metabolites. Similarly, the expression of p53, caspase-1 and -3, and the ratio between Bcl-2/Bax, were all significantly increased at the end of 24 h after exposure to radiation, and reverted to near normal by 48 h, especially those that were treated with GLA (GLA + IR; see Figure 8). These results are in support of the pro-inflammatory and pro-apoptotic actions of radiation and the significant anti-inflammatory and cytoprotective actions of GLA. To confirm these in vitro results, we next performed in vivo studies on C57BL/6J mice.

The most important observation of the present study is the significant increase in the survival of female mice treated with GLA + a lethal dose of radiation (only ~20% survival following whole-body radiation compared to ~80% in GLA-treated animals, see Figure 9 for the protocol of the study and Figure 10 for the survival of animals in various groups). The improvement in survival was accompanied by a substantial increase in body weight in GLA-treated animals. (Figure 10B). This remarkable improvement in survival implies that GLA administration not only suppresses radiation-induced inflammatory events, but also enhances the resolution of inflammation and tissue regeneration. 

It is well known that the gut is one of the target organs of damage by radiation. This is in part due to the high turnover of intestinal epithelial cells. As a result of radiation-induced gastrointestinal damage, patients develop nausea, vomiting, diarrhea, digestion, and nutrient absorption issues that lead to significant weight loss [37,38,39]. Hence, in the present study, we evaluated the potential mechanisms involved in the beneficial action(s) of GLA by measuring various PUFA metabolites and pro and anti-inflammatory cytokines; genes associated with PUFA metabolism, inflammation, and apoptotic pathways; protein expression of Bcl-2, Bax, Nf-κB, IκB, 5-LOX, and COX-2; and finally, the histopathology of the duodenum. These studies were performed on the duodenum tissue of the various groups of animals on days 1, 3, 7, and 14 of the study. 

It is evident from the data shown in Figure 12 that radiation induced a decrease in delta-6-desaturase activity on days 1 and 7, but enhanced its activity on day 3, and by day 14, reached the control value. Delta-6-desatuarse is needed for the conversion of dietary LA and ALA to their long-chain metabolites (see Figure 1A) GLA and stearidonic acid, respectively. This may explain why radiation is able to induce its adverse actions, since radioprotective GLA formation is inhibited. The increase in deleta-6-desaturase on day 3, and its restoration to near normal by day 14, could be a defensive response on the part of the radiation-exposed tissues to generate adequate amounts of GLA from dietary LA to protect themselves from the cytotoxic actions of radiation. The GLA + IR group also showed similar changes in the dynamics of delta-6-desaturase activity with the activity of the enzyme (delta-6-desaturase), which was much higher on all days studied. This suggests that GLA treatment restores the activity of delta-6-desaturase to near normal by days 7 and 14, which may explain the beneficial actions of GLA against lethal radiation (see Figure 10 and Figure 12). The significance of increased GLA formation following its supplementation lies in the fact that it (GLA) has potent anti-inflammatory actions and is the precursor of AA, from which several pro- and anti-inflammatory eicosanoids can be formed (see Figure 1A,B). This argument is supported by the observation that corollary changes in the activities of COX-2 and 5-LOX occurred. For instance, COX-2 showed almost identical changes compared to delta-6-desaturase, whereas 5-LOX showed diametrically opposite changes (see Figure 12). When these changes in COX-2 and 5-LOX are compared to the alterations in the levels of PGE2, LTE4, and LXA4, it is seen that radiation induced a sustained elevation in the concentrations of PGE2 (day 3 > day 1 ≥ day 7 ≥ day 14) with a much-decreased level of LTE4 on all the days. In contrast, LXA4 levels were significantly increased on all days, with a peak on day 1 that showed a gradual and sustained decrease on days 3, 7, and 14 (day 1 > day 3 > day 7 > day 14) (see Figure 13). When these changes in PGE2, LTE4, and LXA4 are compared to those seen in the GLA + IR group, it is evident that PGE2 levels were close to control levels on all days, whereas LTE4 levels were significantly higher on day 1, day 3, and day 7 (day 3 > day 1 ≥ day 7), and by day 14, were lower compared to the control; conversely, LXA4 levels were much lower compared to those seen in the radiation group on all days (Figure 13). The changes in PGE2, LTE4, and LXA4 in the GLA alone group resembled those seen in the GLA + IR group, except that they were much lower in magnitude. Thus, GLA treatment (both in GLA alone and GLA + IR groups)-induced changes in the levels of PGE2, LTE4, and LXA4 are much closer to those seen in the control group, implying that GLA tends to normalize altered eicosanoid levels. Since GLA enhanced the survival of animals treated with lethal radiation, the changes observed in PGE2, LTE4, and LXA4 in all the studied groups seem to suggest that a sustained elevation in LTE4, a muted change in PGE2 (almost no change except slight increase on day 3), and a moderate increase in LXA4 levels, is needed to enhance the survival of animals, and to ensure tissue repair and regeneration of damaged tissues, exposed to lethal radiation (see Figure 13). It is noteworthy that changes in the levels of PGE2, LTE4, and LXA4 seen in GLA alone and GLA + IR groups showed similar pattern. The significant increase in PGE2, muted changes in LTE4, and high levels of LXA4, in response to radiation seem correspond to increased lethality. Extrapolating these changes in PGE2, LTE4, and LXA4 to the changes in COX-2 and 5-LOX enzymes, it is seen that, on days 1 and 3, COX-2 activity was elevated, whereas 5-LOX activity decreased on days 7 and 14; COX-2 was decreased, and 5-LOX increased in the radiation group, and these changes reverted to near normal due to GLA treatment. Thus, there were parallel changes in the activities of COX-2 and 5-LOX enzymes that reflected changes in PGE2 and LTE4 levels (see Figure 12 and Figure 13). These results are in line with the previous data, which showed improved survival of mice by prior administration of 16,16-Dimethyl prostaglandin E2 (DiPGE2), a stable analog of PGE2, against lethal doses ionizing radiation [20,21]. Similar radioprotection was also reported with leukotriene C4 (LTC4) [22]. Based on these results [20,21,22] and the results of the present study, it can be educed that LTE4 is more potent compared to PGE2 to protect animals against lethal radiation. It is possible that both PGE2 and LTE4 are needed for radiation protection, but LTE4 seems to be more potent than PGE2 in this endeavor. The significant increase in the levels of LXA4 following radiation, and its suppression in GLA-treated groups, suggests that an increase in LXA4 immediately following radiation is not desirable. Since LXA4 is a potent anti-inflammatory compound derived from AA, and the association of its increased levels with radiation-induced mortality implies that following radiation exposure (and for that matter any inflammatory stimuli) initial optimal inflammation is needed to derive much needed resolution of inflammation at a subsequent stage. In other words, initial optimal inflammation is needed to initiate long-lasting and sustained resolution of inflammatory events to restore homeostasis. This argument is supported by the high levels of LTE4, with much smaller increase in PGE2, and a moderate but sustained increase in LXA4 (on days 1 > 3 > 7) concentrations observed in GLA + IR-treated animals, which showed a gradual decrease, and almost normal values by day 14 (see Figure 13). Thus, it is important to maintain a dynamic equilibrium among PGE2, LTE4, and LXA4 levels (and similar dynamic equilibrium may exist among TNF, MIF, IL-6, and IL-10; desaturases, COX-2 and LOX enzymes; NF-kB and IkB; and BCL-2 and BAX genes; see Figure 11, Figure 12, Figure 13, Figure 14, Figure 15 and Figure 16) to produce optimum inflammation and trigger the resolution of inflammation in a timely and sustainable fashion. 

This interpretation is supported by the changes in the levels of HMGB1, TNF-α, IL-6, and IL-10 seen in the present study. The results provided in Figure 11 surprisingly did not show any dramatic increases in the levels of both IL-6 and TNF-α compared to the control, including the radiation group. The reason for this lack of response in the levels of both IL-6 and TNF-α (see Figure 11B,C) in response to pro-inflammatory action of radiation could be attributed to the increased levels of PGE2 and LXA4 (see Figure 13), which are known to suppress the production of these cytokines [60,71,72,73,74,75,76,77,78,79,80,81,82,83,84,85,86]. However, it is noteworthy that some studies showed that PGE2 and increased COX-2 expression enhance IL-6 production, and suggested that the inhibition of COX-2 is critical to suppress IL-6 [78,79,80]. This contrasts with the results of the present study, wherein COX-2 expression was enhanced yet IL-6 levels did not show the same degree of increase (see Figure 11, Figure 12 and Figure 13). This lack of increase in IL-6, especially in the radiation group, despite increases in COX-2 expression and enhanced PGE2 levels, may reside in some other event. LXA4 is a potent inhibitor of PGE2, LTE4, IL-6, and TNF-α production, and suppresses COX-2 expression [81,82,83,84]. Since LXA4 levels were significantly increased in all the study groups, especially in radiation alone and GLA + IR groups, from day 1 to day 14 (see Figure 13), it is suggested that it (LXA4) is responsible for the changes in the levels of IL-6, TNF-α, and PGE2, and the alterations in the expression of COX-2 observed. It is noteworthy that enhanced COX-2 expression, and the levels of PGE2, LTE4, and IL-6, are predominantly on day 3 in the radiation alone group, the day on which LXA4 levels started to decline. This implies that LXA4 is the major force that negatively regulates all the pro-inflammatory events (see Figure 11, Figure 12 and Figure 13). It may also be noted here that on day 1, the levels of LXA4 were maximum, which corresponded to the lowest levels of PGE2, LTE4, IL-6, TNF-α, IL-10, and COX-2 expression, especially in the radiation alone group, once again emphasizing the regulatory and potent anti-inflammatory action of LXA4. In addition, it has not escaped our attention that 5-LOX expression and HMGB1 levels were maximal when the levels of LXA4 were highest on day 1 (see Figure 11, Figure 12 and Figure 13). These results are interpreted to mean that HMGB1 is the best predictor of inflammation, and 5-LOX expression corresponds to the increased generation of LXA4, as seen in the present study. It is interesting to note that the levels of IL-10 were maximum on day 3, and started to slowly decline on days 7 and 14 in the GLA + IR group (the group in which survival against lethal radiation was dramatically increased as a result of GLA treatment, see Figure 10), whereas in the radiation alone group, its (IL-10) levels were lowest, with a dramatic increase on day 14, perhaps implying a natural recovery process from radiation effects (see Figure 11). This argument is supported by the reports that LXA4 improves survival and recovery from various pro-inflammatory insults, which is, at least in part, due to the enhanced formation of IL-10 [84,85,86]. These results emphasize the anti-inflammatory action of GLA by its ability to modulate cytokines, eicosanoids, and genes concerned with their formation (such as Delta-6-desaturase, COX-2, and 5-LOX).

The anti-inflammatory action of GLA is further attested by the changes in the expression of NF-kB and IkB, and BCL-2 and BAX genes (see Figure 14, Figure 15 and Figure 16). Radiation induced significant increases in the expression of NF-kB and BAX, and their proteins, which reverted to normal in the GLA + IR group, implying the anti-inflammatory and pro-survival actions of GLA. Similarly, even the altered expressions of COX-2 and 5-LOX also reverted to near normal in the GLA + IR group, attesting to the anti-inflammatory and pro-survival action of GLA. These results are supported by the histopathological studies performed on the duodena of various groups (Figure 17), which showed that GLA + IR-treated animals had less cell/tissue damage, implying enhanced repair from radiation-induced injury. It is noteworthy that even alterations, induced by radiation, in antioxidant/pro-oxidant levels in the duodenum were restored to near normal by GLA treatment, as is evident from the results shown in Table 3.

## 5. Conclusions

The results of the present study and previous data [2,3,4,5,6,7,8,20,21,22,23,24,25] are in line with the idea that bioactive lipids, especially unsaturated fatty acids (GLA, DGLA, AA, EPA, and DHA), and their metabolites (both pro- and anti-inflammatory products) play a significant role in the biological actions of radiation and radiation-induced injury, and its resolution. The results of the present study suggest that both inflammatory events and anti-inflammatory events occur simultaneously (see Figure 11, Figure 12, Figure 13, Figure 14, Figure 15 and Figure 16), with a dynamic equilibrium among various components of these processes. It is evident from the data in Figure 11, Figure 12, Figure 13, Figure 14, Figure 15 and Figure 16 that a sufficient degree of inflammation needs to occur first so that the appropriate inflammation resolution events are engaged to ensure the resolution of inflammation and wound healing. This may explain why, within 24–72 h of exposure to radiation, there is a dramatic increase in the production of PGE2 and LTE4, and in the expression of COX-2, increased generation of HMGB1 vs. IL-10, and an increase in the ratio of NF-kB/IkB and BCL-2/Bax (See Figure 11, Figure 12, Figure 13, Figure 14, Figure 15 and Figure 16), which corresponded to the histopathological changes seen in the duodenum of radiation-treated animals (Figure 17). Once these acute inflammatory changes have occurred, they were followed by equally dramatic opposite changes (decrease in PGE2, LTE4, HMGB1 concentrations; NF-kB/IkB and BCL-2/BAX and COX-2 and 5-LOX expressions). The initial increase in LXA4 levels seen in radiation-exposed animals started decreasing by 72 h, implying that a continued increase in its levels is not needed for anti-inflammatory events to set in, and for wound healing to occur. These dramatic and dynamic, and closely coordinated changes in the concentration and expression of molecules and genes concerned with inflammation and its resolution are reflected in the histopathological changes seen in the duodenum of these animals. This explanation may explain the beneficial actions of PGE2 and LTC4 previously reported [20,21,22,23,24,25]. These results are further supported by the observation that PGE2 and PGI2 enhance tissue regeneration [26,27,87]. Even though LXA4 is needed for the resolution of inflammation and wound healing, its production needs to occur at an appropriate time. Ill-timed production of LXA4 could be hazardous, as evident from our results, which show that LXA4 administration prior to radiation exposure did not enhance survival of animals (see Figure 10E,F). 

There are some surprises in the results obtained in the present study. For instance, GLA was found to be effective in preventing mortality due to radiation, while DGLA, AA, EPA, and DHA were ineffective (See Figure 10C,D). DGLA is the precursor of PGE1, an anti-inflammatory molecule, whereas PGE2, PGI2, LTD4, and LXA4 are derived from AA (see Figure 1A,B). Hence, it is reasonable to predict that AA would be the most appropriate fatty acid to prevent radiation-induced mortality. However, this was not so. Previously, it was shown that the administration of AA enhances LXA4 formation with little or no change in PGE2 levels [88,89]. These results suggest that the administration of AA probably augmented LXA4 formation upon exposure to radiation, which resulted in no change in the mortality rate (Figure 10E,F). These results are similar to those seen in the LXA4 + radiation group. GLA can be converted to AA by the action of delta-5-desaturase. In the present study, delta-5-desaturase activity was below the detection limit. However, this does not rule out the presence of minimal activity of delta-5-desaturase, which is sufficient to convert the administered GLA to AA into much-needed PGE2, LTE4, and LXA4, as observed in the present study. 

Another surprising observation is the failure of male mice to show any enhanced survival following GLA administration upon radiation, unlike females. The exact reason for this observation is not clear, and we intend to investigate this in the future.

Past studies have revealed that gut microbiota can improve the survival of experimental animals exposed to lethal radiation [90,91,92,93,94] by virtue of their ability to produce short-chain fatty acids and various tryptophan metabolites. We did not evaluate the alterations in gut microbiota in the present study. Nonetheless, previous studies performed by ourselves, and by others, revealed that various PUFAs (including GLA) have beneficial actions on gut microbiota [95,96,97]. Hence, one distinct possibility by which GLA is beneficial against radiation can be ascribed to its ability to alter gut microbiota (See Figure 18). However, this possibility needs to be verified.

In summary, the results of the present study suggest that GLA is a radioprotective agent that needs to be explored and exploited for use in future human studies.

## Figures and Tables

**Figure 1 biomolecules-12-00797-f001:**
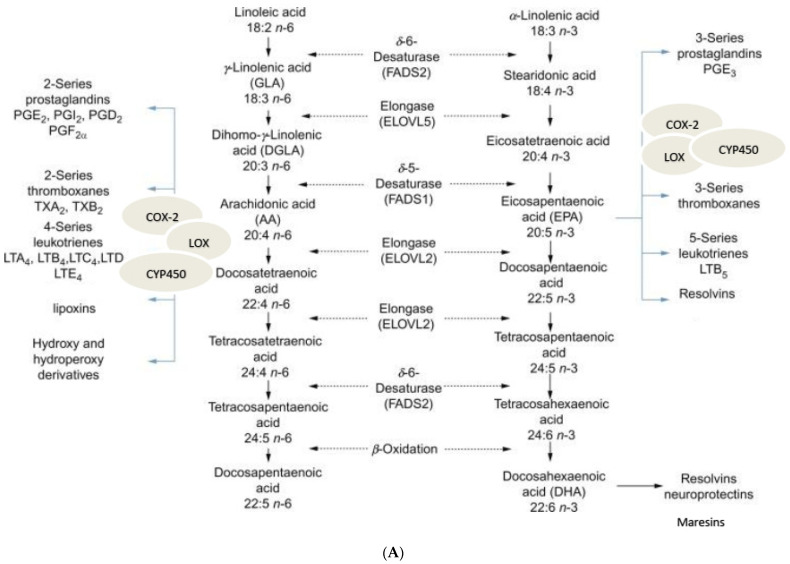
(**A**) Metabolism of essential fatty acids. (**B**) Metabolism of arachidonic acid.

**Figure 2 biomolecules-12-00797-f002:**
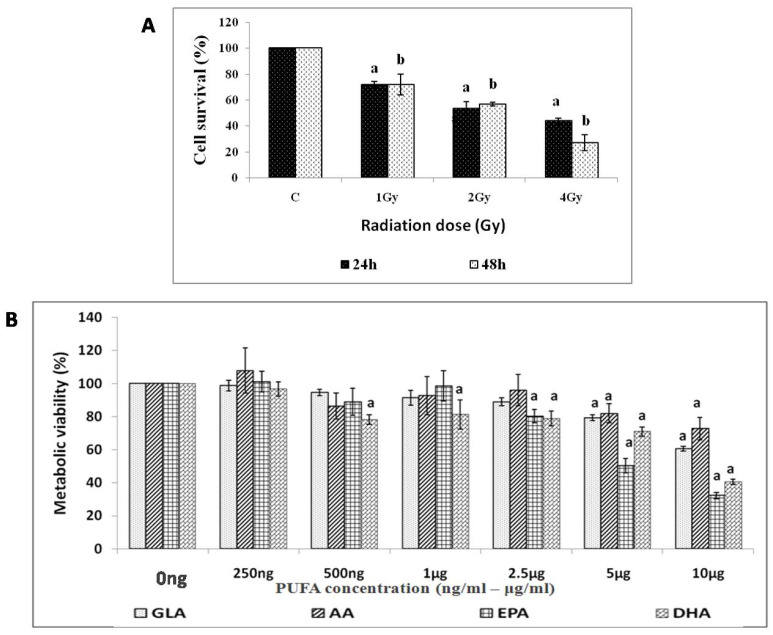
(**A**). Effect of ionizing radiation on the viability of RAW 264.7 cells as assessed by trypan blue dye exclusion method. Values (n = 6) expressed as mean ± SEM. ^a^ (*p* < 0.05) represents significant decrease in %Survival when compared with control at 24 h. ^b^ (*p* < 0.05) represents significant decrease in %Survival when compared with control at 48 h. (**B**). Study of effect of PUFAs, GLA, AA, EPA, and DHA, on the viability of RAW 264.7 cells by MTT assay. Values (n = 6) expressed as mean ± SEM. ^a^ (*p* < 0.05) represents significant decrease in %Survival when compared with control at 24 h. GLA, gamma-linolenic acid; AA, arachidonic acid; EPA, eicosapentaenoic acid; DHA, docosahexaenoic acid.

**Figure 3 biomolecules-12-00797-f003:**
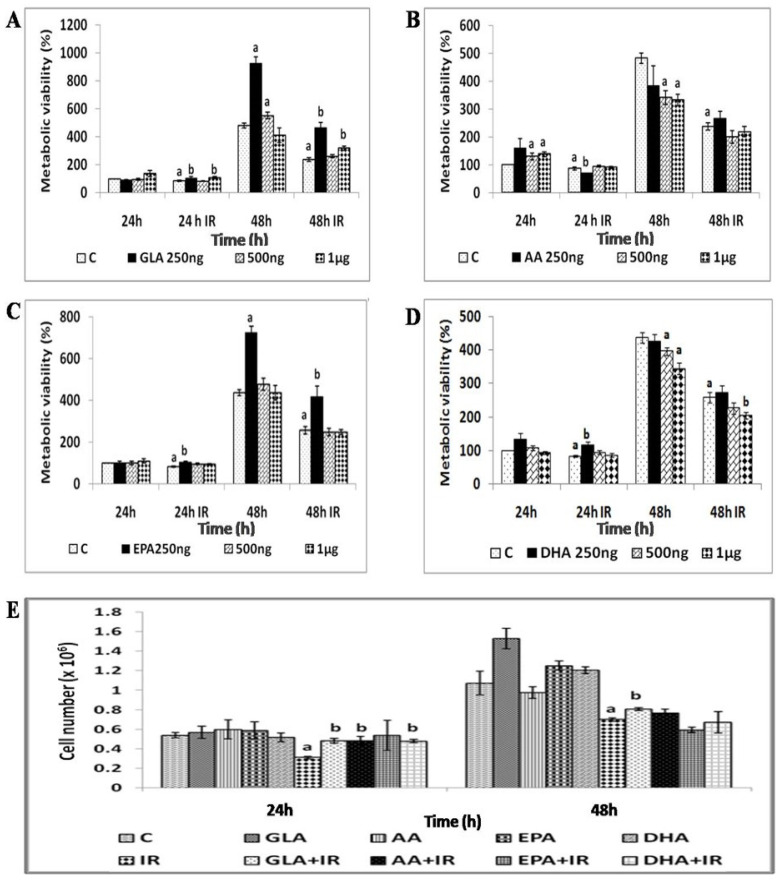
Effect of various doses of different PUFAs on radiation-induced cytotoxic action on RAW 264.7 cells in vitro at 24 and 48 h post-radiation, as assessed by MTT assay. Doses of PUFAs used were 250 ng/mL, 500 ng/mL, and 1 μg/mL). (**A**) = GLA; (**B**) = AA; (**C**) = EPA; and (**D**) = DHA; (**E**) **=** Effect of various PUFAs (GLA, AA, EPA, and DHA) on the growth kinetics of RAW 264.7 cells following irradiation. All values (n = 6) are expressed as mean ± SEM. Significance (*p* < 0.05) represented by ^a^, ^b^ when compared with control and irradiation, respectively. C, control; GLA, gamma-linolenic acid; AA, arachidonic acid; EPA, eicosapentaenoic acid; DHA, docosahexaenoic acid; IR, irradiation (2 Gy).

**Figure 4 biomolecules-12-00797-f004:**
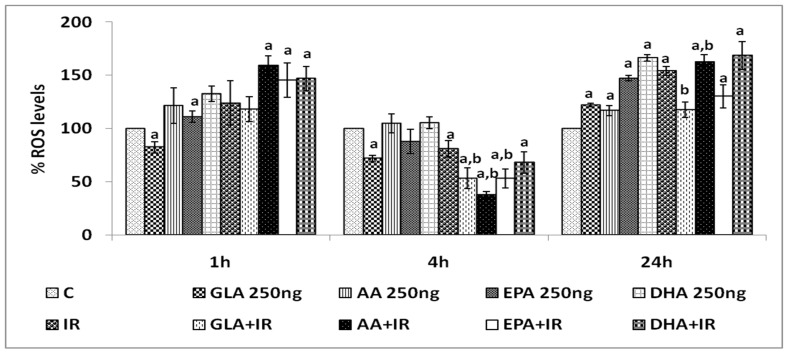
Effect of various PUFAs on radiation-induced reactive oxygen species (ROS) in RAW 264.7 cells using DCFH_2_DA. Values (n = 6) expressed as mean ± SEM. Significance (*p* < 0.001) represented by ^a^, ^b^ when compared with control, PUFAs, and irradiation, respectively. C, control; GLA, gamma-linolenic acid; AA, arachidonic acid; EPA, eicosapentaenoic acid; DHA, docosahexaenoic acid; IR, irradiation.

**Figure 5 biomolecules-12-00797-f005:**
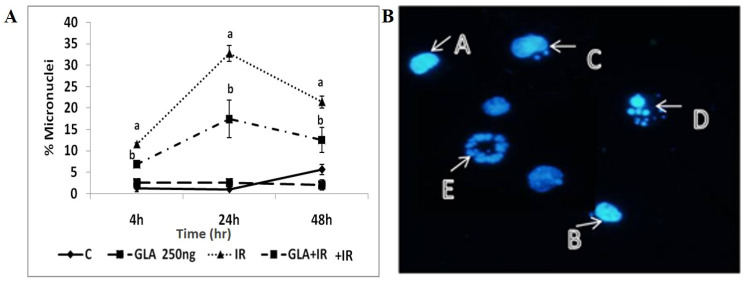
Estimation of DNA damage in RAW 264.7 cells through micronuclei estimation using DAPI (**A**). Values (n = 6) expressed as mean ± SEM. Significance (*p* <0.01) represented by ^a^, ^b^ when compared with control and irradiation, respectively. C, control; GLA, gamma-linolenic acid; IR, irradiation (2 Gy). Pictorial representation of RAW 264.7 cells stained with DAPI under fluorescence microscope (**B**), with nucleus visible at 100× resolution. A. normal nucleus, B. nucleus with single micronucleus, C. nucleus with two micronuclei, D. nucleus with multiple micronuclei, and E. nucleus of apoptotic cell.

**Figure 6 biomolecules-12-00797-f006:**
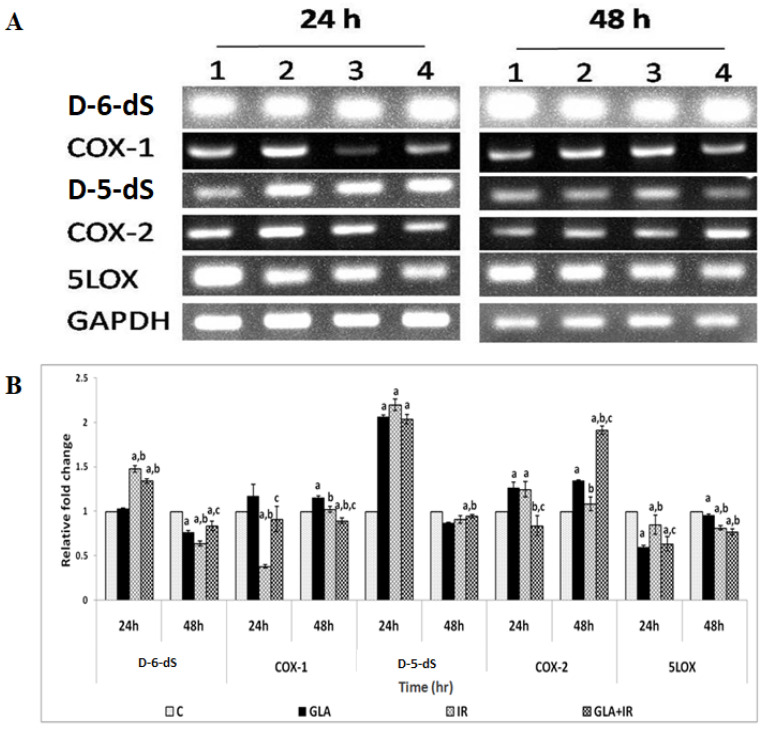
Effect of GLA and irradiation on genes of EFA (essential fatty acid) metabolic pathways (desaturases, COX-1, COX-2, and 5-LOX) in RAW 264.7 cells at 24 and 48 h post-irradiation (**A**,**B**). Values (n = 3) expressed as mean ± SEM. ^a^, ^b^, ^c^ Significant (*p* < 0.01) when compared with control, GLA, and irradiation, respectively. 1, C; 2, GLA; 3, IR; and 4, GLA + IR. C, control; GLA, gamma-linolenic acid; IR, irradiation (2 Gy).

**Figure 7 biomolecules-12-00797-f007:**
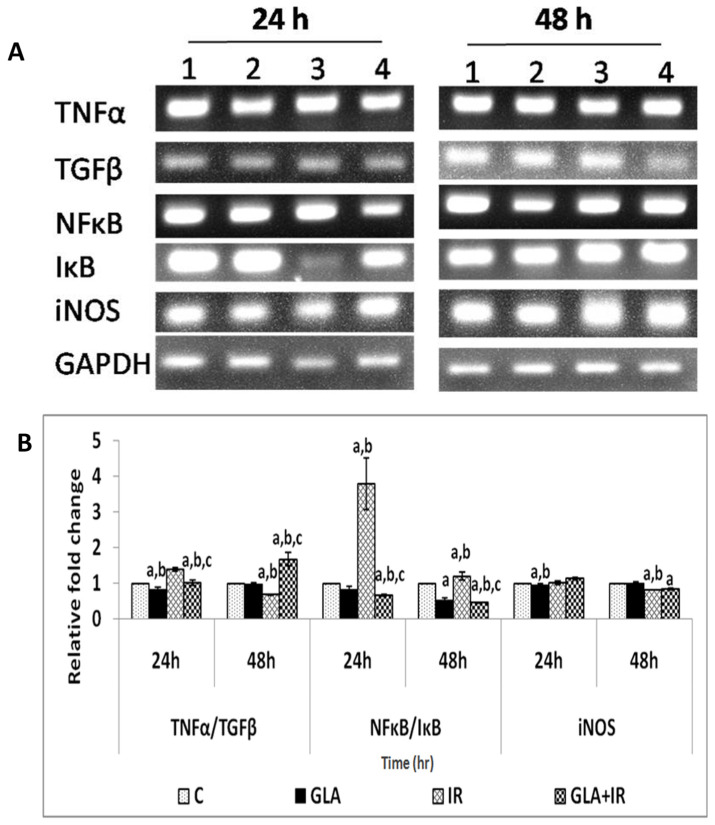
Effect of GLA and IR on genes regulating the inflammatory pathway (TNF, TGF, NF-kB, IkB, iNOS) in RAW 264.7 cells at 24 and 48 h post-irradiation (**A**,**B**). Values (n = 3) expressed as mean ± SEM. ^a^, ^b^, ^c^ Significant (*p* < 0.01) when compared with control, GLA, and irradiation, respectively. 1, C; 2, GLA; 3, IR; and 4, GLA + IR. C, control; GLA, gamma-linolenic acid; IR, irradiation (2 Gy).

**Figure 8 biomolecules-12-00797-f008:**
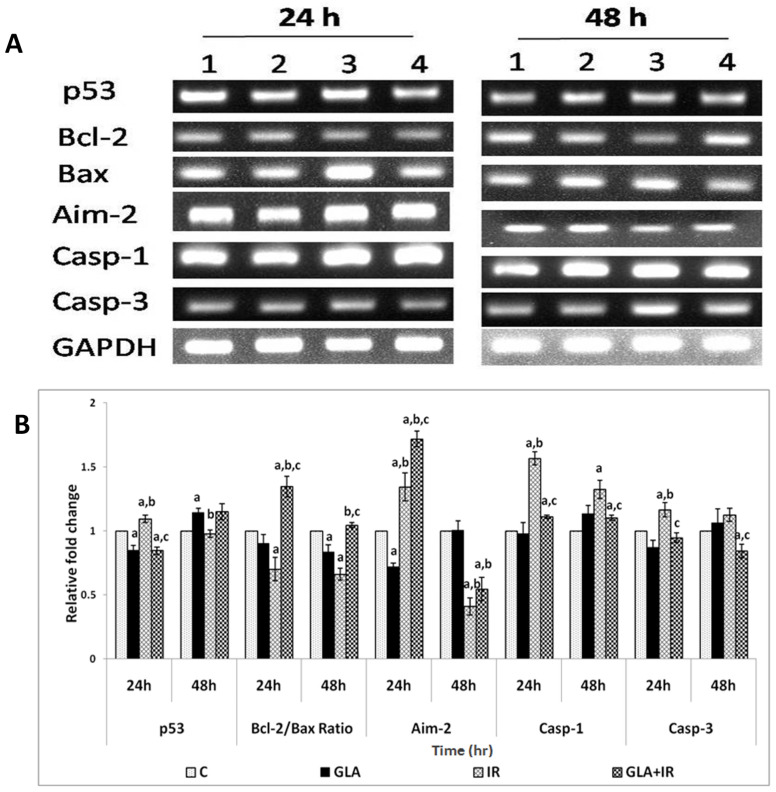
Effect of GLA and irradiation on genes associated with apoptotic pathways in RAW 264.7 cells at 24 and 48 h post-irradiation (**A**,**B**). Values (n = 3) expressed as mean ± SEM. ^a^, ^b^, ^c^ Significant (*p* < 0.01) when compared with control, GLA, and irradiation, respectively. 1, C; 2, GLA; 3, IR; and 4, GLA + IR. C, control; GLA, gamma-linolenic acid; IR, irradiation (2 Gy).

**Figure 9 biomolecules-12-00797-f009:**
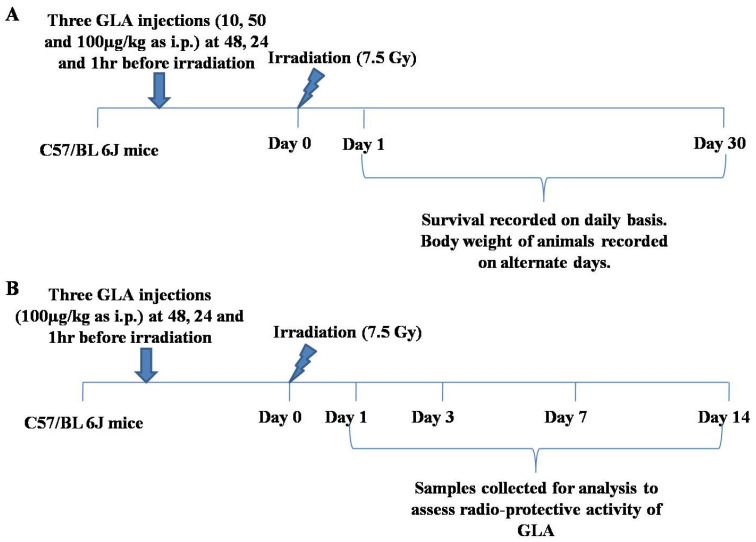
(**A**). **Protocol of the in vivo study with radiation and GLA treatment.** Mice were pre-acclimatized and treated with GLA (10, 50, and 100 μg/kg body weight at 48, 24, and 1 h prior to irradiation (7.5 Gy) at a dose rate of 1 Gy/min. (**B**). **Protocol for mechanistic studies.** Mice were pre-acclimatized and treated with GLA (100 μg/kg body weight at 48, 24, and 1 h prior to irradiation (7.5 Gy) at a dose rate of 1 Gy/min. Samples were collected from mice on days 1, 3, 7, and 14 for further analysis.

**Figure 10 biomolecules-12-00797-f010:**
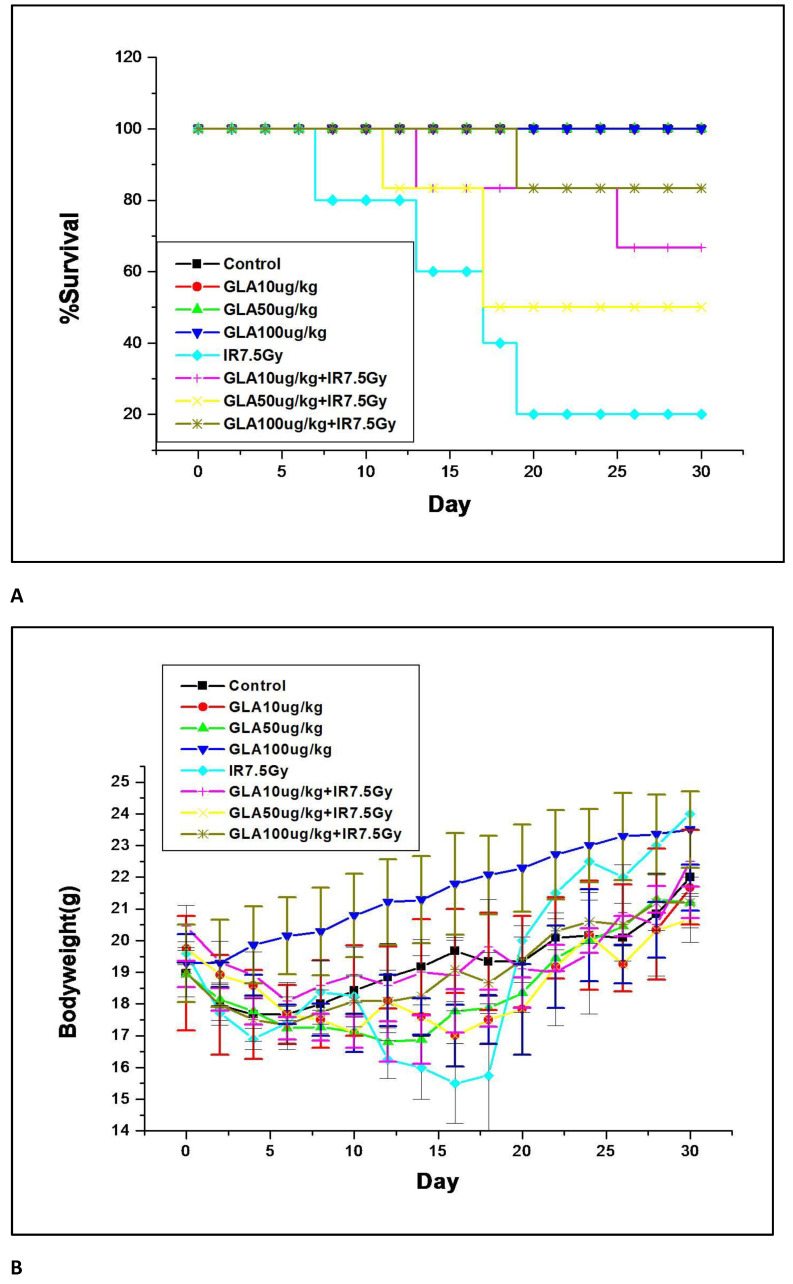
Effect of various PUFAS and LXA4 on radiation-induced mortality in female mice. Mice were pre-treated with GLA (10, 50, and 100 μg/kg) at 48, 24, and 1 h prior to irradiation and were subjected to total-body irradiation of 7.5 Gy (at 1 Gy/min). Values (n = 12) expressed as %Survival (**A**) and variation in body weight (**B**). Mice were pre-treated with 100 μg/kg PUFA (DGLA, AA, EPA, and DHA) at 48, 24, and 1 h prior to irradiation, and were subjected to total-body irradiation of 7.5 Gy (at 1 Gy/min). Values (n = 6) expressed as %Survival (**C**) and variation in body weight (**D**). Mice were pre-treated with 10, 50, 100 ng/kg of LXA4 at 48, 24, and 1 h prior to irradiation, and were subjected to total-body irradiation of 7.5 Gy (at 1 Gy/min). Values (n = 6) expressed aISurvival (**E**). %Survival with LXA4 50 ng/kg post-treatment (**F**) on day 7, 14, 21, and 28, respectively. DGLA, di-homo-gamma-linolenic acid; AA, arachidonic acid; EPA, eicosapentaenoic acid; DHA, docosahexaenoic acid; IR, irradiation; LXA4, lipoxin A4.

**Figure 11 biomolecules-12-00797-f011:**
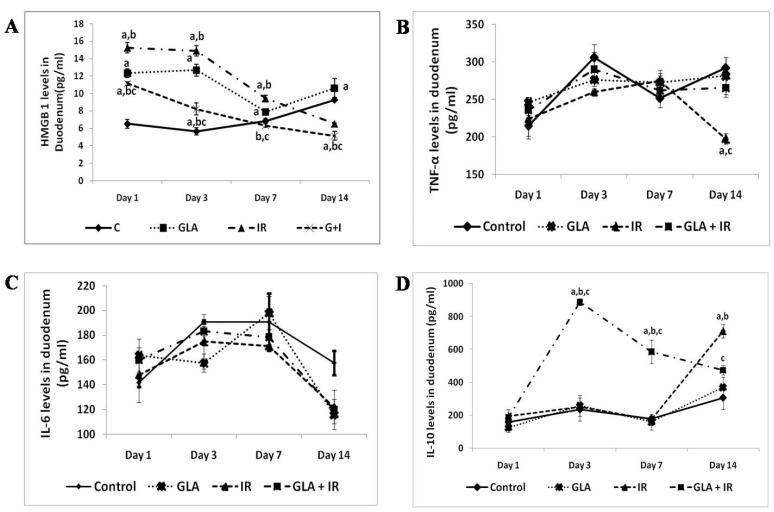
Estimation of duodenal concentrations of cytokines HMGB1 (**A**), TNF-α (**B**), IL-6 (**C**), and IL-10 (**D**) by ELISA method. Mice were pre-treated with 100 μg/kg GLA at 48, 24, and 1 h prior to irradiation, and were subjected to total-body irradiation of 7.5 Gy (at 1 Gy/min). Duodenum samples were obtained on day 1, day 3, day 7, and day 14 post-irradiation. All values are expressed as mean ± SEM (n = 3). Statistical significance was calculated using *t*-tests, and *p*-values < 0.05 are represented by ^a^, ^b^, ^c^ when compared to control, GLA, and irradiation, respectively. C, control; GLA, gamma-linolenic acid; IR, irradiation (7.5 Gy).

**Figure 12 biomolecules-12-00797-f012:**
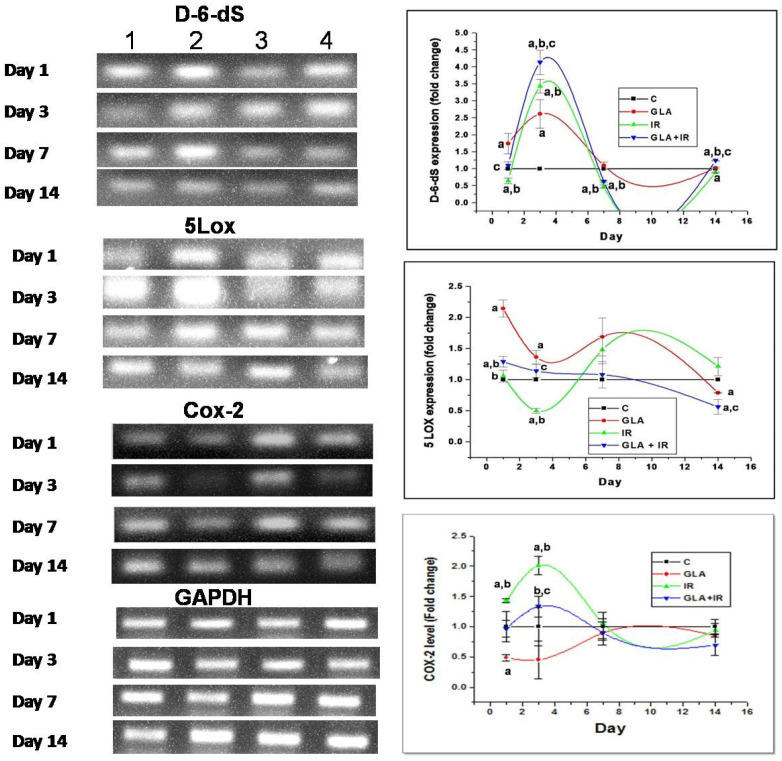
Effect of GLA and irradiation on genes associated with PUFA metabolism in duodenum tissue on day 1, day 3, day 7, and day 14. Values (n = 3) expressed as mean ± SEM. ^a^, ^b^, ^c^ Significant (*p* < 0.01) when compared with control, GLA, and irradiation, respectively. 1, C; 2, GLA; 3, IR; and 4, GLA + IR. C, control; GLA, gamma-linolenic acid; IR, irradiation (7.5 Gy).

**Figure 13 biomolecules-12-00797-f013:**
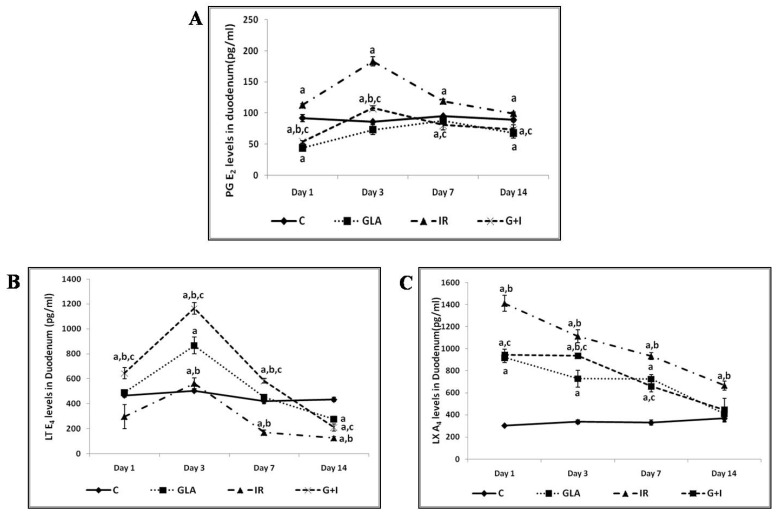
Estimation of PUFA metabolite PG E_2_ (**A**), LT E_4_ (**B**), and LXA4 (**C**) levels by ELISA method. Mice were pre-treated with 100 μg/kg GLA at 48, 24, and 1 h prior to irradiation, and were subjected to total-body irradiation of 7.5 Gy (at 1 Gy/min). Duodenum samples were obtained on day 1, day 3, day 7, and day 14 post-irradiation. All the values are expressed as mean ± SEM (n = 3). Statistical significance was calculated using *t*-tests, and *p*-values < 0.001 are represented by ^a^, ^b^, ^c^ when compared to control, GLA, and IR, respectively. C, control; GLA, gamma-linolenic acid; IR, irradiation (7.5 Gy).

**Figure 14 biomolecules-12-00797-f014:**
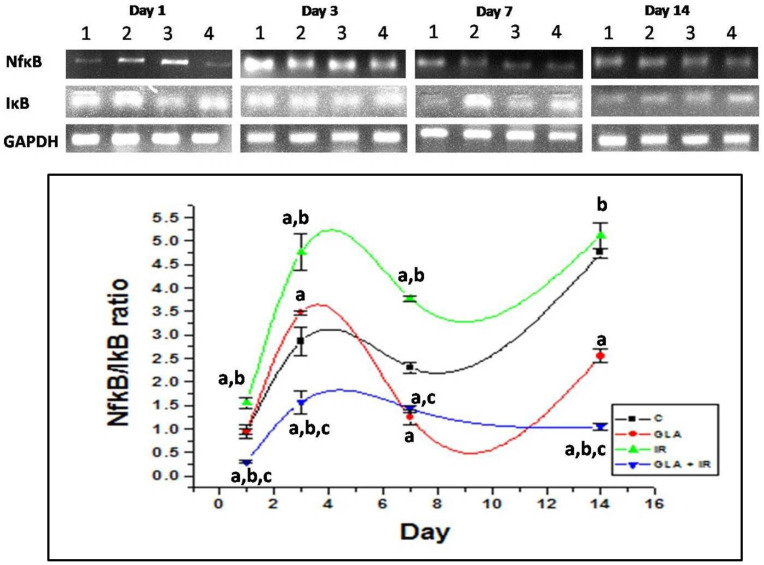
**Effect of GLA and radiation on the expression of NF-kB/IkB in duodenal tissue of mice on days 1, 3, 7, and 14.** Values (n = 3) are expressed as mean ± SEM. ^a^, ^b^, ^c^ Significant (*p* < 0.01) when compared with control, GLA, and IR, respectively. 1, C; 2, GLA; 3, IR; and 4, GLA + IR. C, control; GLA, gamma-linolenic acid; IR, irradiation (7.5 Gy).

**Figure 15 biomolecules-12-00797-f015:**
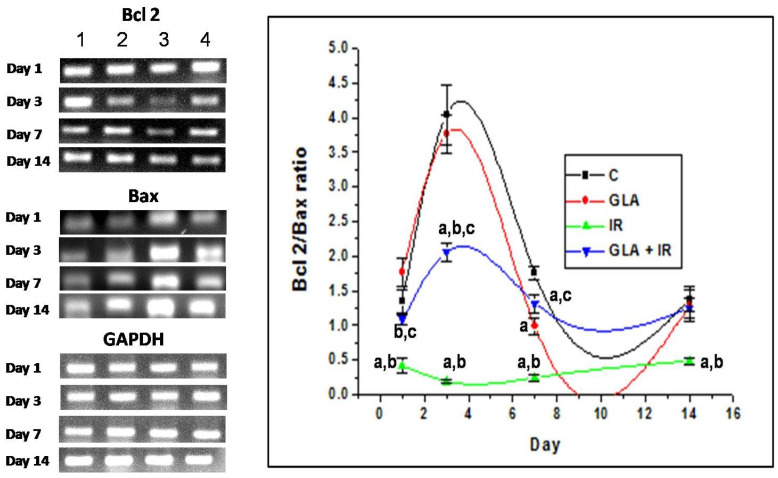
Effect of GLA and irradiation on genes associated with apoptotic pathways (BCL-2 and Bax) in duodenum tissue on day 1, day 3, day 7, and day 14. Values (n = 3) expressed as mean ± SEM. ^a^, ^b^, ^c^ Significant (*p* < 0.01) when compared with control, GLA, and IR, respectively. 1, C; 2, GLA; 3, IR; and 4, GLA + IR. C, control; GLA, gamma-linolenic acid; IR, irradiation (7.5 Gy).

**Figure 16 biomolecules-12-00797-f016:**
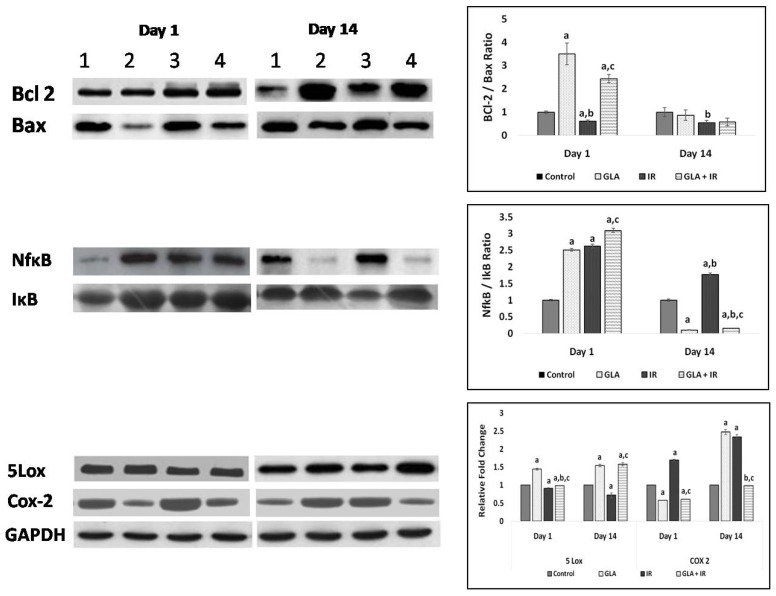
Effect of GLA and irradiation on protein expression (Bcl-2, Bax, Nf-κB, IκB, 5-LOX, and COX-2) in duodenum tissue on day 1 and day 14 post-irradiation by Western blotting. GAPDH was used as a loading control. Values (n = 3) expressed as mean ± SEM. ^a^, ^b^, ^c^ Significant (*p* < 0.05) when compared with control, GLA, and irradiation, respectively. 1, C; 2, GLA; 3, IR; and 4, GLA + IR. C, control; GLA, gamma-linolenic acid; IR, irradiation (7.5 Gy).

**Figure 17 biomolecules-12-00797-f017:**
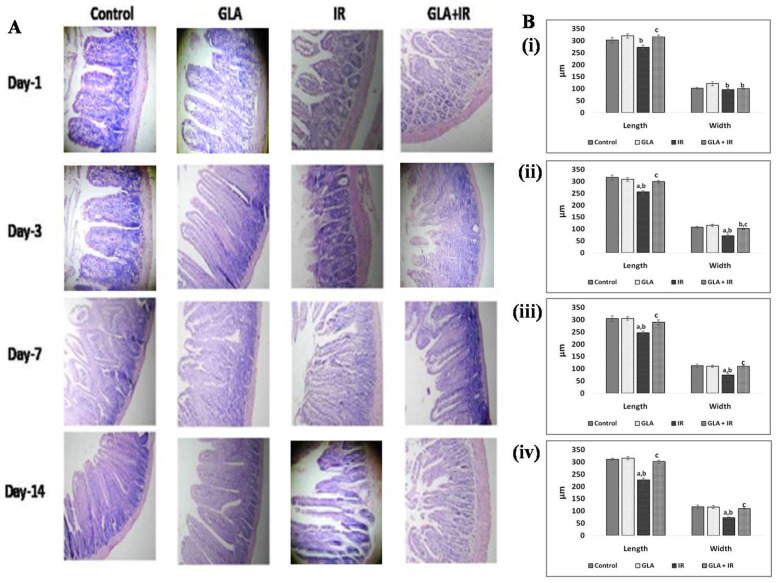
Histology of duodenum of surviving mice (**A**). Control, non-irradiated mice; GLA, mice treated with 100 μg/kg GLA; IR, irradiated mice (7.5 Gy); GLA + IR, GLA-pre-treated and irradiated mice. On day 1, day 3, day 7, and day 14, cells were stained with hematoxylin and eosin (10× magnification). Villi length and width (in μm) were measured (**B**). Values (n = 3) expressed as mean ± SEM. ^a^, ^b^, ^c^ Significant (*p* < 0.05) when compared with control, GLA, and irradiation, respectively. GLA, gamma-linolenic acid; IR, irradiation (7.5 Gy).

**Figure 18 biomolecules-12-00797-f018:**
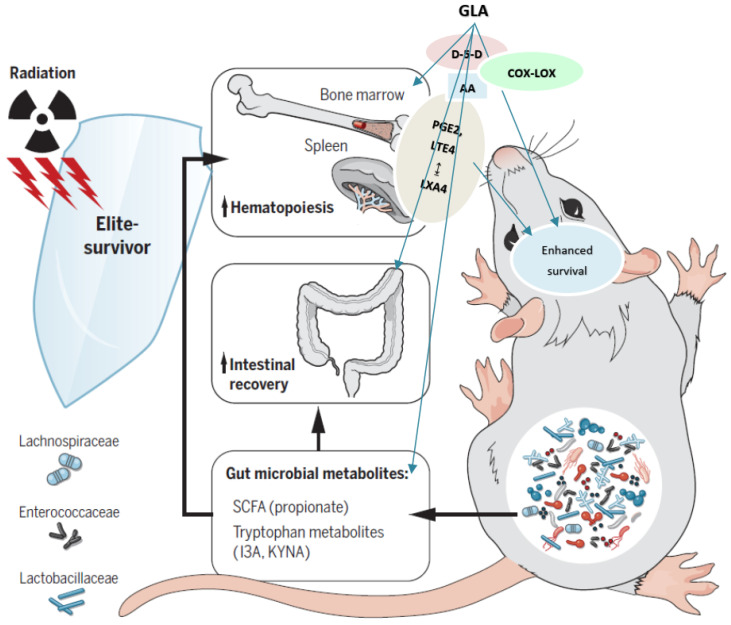
Scheme showing potential mechanisms involved in the beneficial actions of GLA and its products (PGE2/LTE4/LXA4 derived from AA and AA, in turn, is derived from GLA by the action of delta-5-desaturase).

**Table 1 biomolecules-12-00797-t001:** Details of primer product size and their respective annealing temperatures used in semi-quantitative PCR.

Gene	Primer	Sequence (5′-3′)	Annealing Temp (°C)	Product Size (bp)
p53	F	GCTGGTTCATCACTCCTCCC	58.4	216
	R	GCTTCCCCATTTCACTCTGG		
Bcl-2	F	CTCGTCGCTACCGTCGTGACTTCG	66	242
	R	CAGATGCCGGTTCAGGTACTCAGTC		
Bax	F	CGGCGAATTGGAGATGAACTG	58	161
	R	GCAAAGTAGAAGAGGGCAACC		
AIM-2	F	ACAAAGTGCGAGGAAGGAGA	53.7	125
	R	TTTGGCTTTGCAGCCTTAAT		
Caspase-1	F	GAGAAGAGAGTGCTGAATCAG	55.5	400
	R	CAAGACGTGTACGAGTGGTTG		
Caspase-3	F	TGTCATCTCGCTCTGGTACG	55.5	392
	R	AGCCTCCACCGGTATCTTCT		
TNF-α	F	CTGAACTTCGGGGTGATCGG	58.4	122
	R	GGCTTGTCACTCGAATTTTGAGA		
TGF-β	F	GGATACCAACTATTGCTTCAGCTCC	65	156
	R	AGGCTCCAAATATAGGGGCAGGGTC		
NF-κB	F	CAGCACTGATGGCATGGGGGACACTGACA	68	588
	R	CCCAATGCATAGCCATTACACGTTT TTCACCTTAAATCTGCTT		
IκB	F	CTTGGTGACTTTGGGTGCTGAT	59	101
	R	GCGAAACCAGGTCAGGATTC		
iNOS	F	CCCTTCCGAAGTTTCTGGCAGCAGC	60	499
	R	GGCTGTCAGAGAGCCT CGTGGCTTTGG		
COX-1	F	AGGAGATGGCTGCTGAGTTGG	57.5	601
	R	AATCTGACTTTCTGAGTTGCC		
COX-2	F	ACACACTCTATCACTGGCACC	57.5	274
	R	TTCAGGGAGAAGCGTTTGC		
D-6-dS	F	TGCCTGGGTCATCCTCTCGTA	58	58
	R	GGCTGTGACGAGGGTAGGAA		
D-5-dS	F	TGTGTGGGTGACACAGATGA	53.5	115
	R	GTTGAAGGCTGATTGGTGAA		
5-LOX	F	ATGTTGGCATCTAGGTGCAGTGTG	62	114
	R	ATCATGGCTTCCTTCACTGGCTTC		
GAPDH	F	AACTTTGGCATTGTGGAAGG	54	223
	R	ACACATTGGGGGTAGGAACA		

**Table 2 biomolecules-12-00797-t002:** Estimation of antioxidant status in RAW 264.7 cells. RAW 264.7 cells were pre-treated with 250 ng/mL GLA 24 h prior to irradiation at 2 Gy (at 1 Gy/min). Supernatant and cell lysate were obtained at 24 and 48 h post-radiation, and were used for the estimation of various antioxidants. Values (n = 6) expressed as mean ± SE. Statistical significance was determined by *t*-test (*p* < 0.001). Significance expressed as ^a^, ^b^, and ^c^ significance with control, GLA, and IR of same time point, respectively. C, control; GLA, gamma-linolenic acid; IR, irradiation (2 Gy).

Time	Group	LPO/NO Ratio	SOD (Units/mg ptn)	Catalase (µM H2O2/min/g ptn)	GST (µM/min/g ptn)	GPX (µM/min/g ptn)
**24 h**	**Control**	0.042 ± 0.004	198.68 ± 26.62	12,189.58 ± 1374.44	12.12 ± 2.96	93,868.97 ± 11,205.50
	**GLA 250 ng**	0.05 ± 0.006	65.83 ± 9.10 ^a^	14,755.81 ± 1295.06	33.18 ± 0.69 ^a^	56,963.98 ± 2681.21
	**IR 2 Gy**	0.058 ± 0.006 ^a^	22.38 ± 2.04 ^a,b^	2318.51 ± 252.59 ^a,b^	2.45 ± 0.13 ^b^	11,870.93 ±477.72 ^a,b^
	**GLA + IR**	0.043 ± 0.004 ^c^	44.71 ± 5.95 ^a,c^	9107.89 ± 1237.38 ^b,c^	3.83 ± 0.57 ^b^	30,631.42 ± 1366.69 ^a,b,c^
**48 h**	**Control**	0.056 ± 0.007	96.63.84 ± 15.95	2013.84 ± 3079.58	17.07 ± 1.14	94,664.32 ± 11,809.67
	**GLA 250 ng**	0.079 ± 0.006	38.24 ± 10.19	20,689.78 ± 1258.46	30.45 ± 2.26 ^a^	52,692.46 ± 1713.91
	**IR 2 Gy**	0.222 ± 0.013 ^a,b^	6.9 ± 1.6 ^a,b^	859.86 ± 61.84 ^a,b^	3.34 ± 0.34 ^a,b^	10,702.34 ± 391.29 ^b,c^
	**GLA + IR**	0.085 ± 0.012^c^	28.54 ± 2.20 ^a,c^	6322.51 ± 1398.04 ^a,b,c^	13.57 ± 1.13 ^b,c^	28,497.30 ± 1218.81 ^a,b,c^

**Table 3 biomolecules-12-00797-t003:** Mouse duodenal tissue NO, lipid peroxides, and antioxidants following various treatments.

	Group	Nitric Oxide (µM)	Lipid Peroxide (µM)	NO/LPO Ratio	SOD (Units/g Protein)	Catalase (µM H2O2/min/g Protein)	GST (µM/min/g Protein)	GPX (µM/min/g Protein)
Day 1	Control	1.92 ± 0.07	1.26 ± 0.06	1.52 ± 0.02	5.44 ± 0.81	853.02 ± 61.10	10.87 ± 0.50	29.19 ± 0.59
	GLA 100 µg	2.56 ± 0.39	1.05 ± 0.10	2.72 ± 0.63	7.51 ± 1.35	804.58 ± 119.91	12.37 ± 1.48	43.63 ± 4.31 ^a^
	IR 7.5 Gy	2.70 ± 0.46	1.38 ± 0.29	2.25 ± 0.34	10.41 ± 2.17	944.28 ± 98.88	10.43 ± 0.84	32.87 ± 4.03
	GLA + IR	3.41 ± 0.43 ^a^	1.44 ± 0.4	3.33 ± 0.94 ^a^	13.11 ± 1.55 ^a^	2052.89 ± 544.06 ^a,b,c^	7.97 ± 1.24 ^b^	44.65 ± 5.19 ^a^
Day 3	Control	3.77 ± 0.61	1.43 ± 0.18	2.96 ± 0.57	6.60 ± 1.30	1613.74 ± 512.70	4.36 ± 0.31	38.48 ± 4.97
	GLA 100 µg	2.47 ± 0.20	1.07 ± 0.08	2.41 ± 0.31	7.93 ± 1.53	1981.82 ± 417.53	6.97 ± 0.30 ^a^	59.04 ± 4.38 ^a^
	IR 7.5 Gy	3.31 ± 0.63	0.88 ± 0.22	5.43 ± 1.72	4.23 ± 1.27	1319.49 ± 78.28	4.08 ± 0.16 ^b^	47.81 ± 1.69 ^b^
	GLA + IR	2.98 ± 0.65	0.66 ± 0.14 ^b^	5.95 ± 1.86	6.64 ± 1.54	2242.91 ± 284.87 ^c^	6.79 ± 0.91 ^a,c^	53.48 ± 6.43
Day 7	Control	4.36 ± 0.39	1.13 ± 0.07	3.94 ± 0.44	55.91 ± 0.84	869.36 ± 53.96	0.49 ± 0.01	33.52 ± 0.78
	GLA 100 µg	5.55 ± 0.06 ^a^	1.27 ± 0.09	4.52 ± 0.45	52.09 ± 0.08 ^a^	1171.99 ± 118.60 ^a^	0.39 ± 0.02 ^a^	28.78 ± 0.58 ^a^
	IR 7.5 Gy	4.20 ± 0.23 ^b^	1.04 ± 0.10	4.19 ± 0.38	35.45 ± 8.70^a^	422.73 ± 124.27 ^a,b^	0.70 ± 0.13 ^b^	38.80 ± 5.59
	GLA + IR	5.79 ± 0.15 ^a,c^	0.17 ± 0.07 ^a,b,c^	77.53 ± 21.07 ^a,b,c^	64.71 ± 2.16 ^a,b,c^	1096.76 ± 259.34 ^a,c^	1.44 ± 0.31 ^a,b,c^	42.80 ± 2.72 ^a,b^
Day 14	Control	2.25 ± 0.34	0.89 ± 0.01	2.53 ± 0.40	19.23 ± 3.54	454.17 ± 82.91	1.18 ± 0.20	9.46 ± 0.24
	GLA 100 µg	7.23 ± 0.09 ^a^	1.98 ± 0.45 ^a^	4.90 ± 1.08	16.37 ± 0.25	739.23 ± 62.85 ^a^	0.99 ± 0.04	8.69 ± 0.93
	IR 7.5 Gy	7.29 ± 0.18 ^a^	0.29 ± 0.12 ^a,b^	235.95 ± 99.33 ^a,b^	8.03 ± 3.07 ^a,b^	469.72 ± 31.52 ^b^	1.43 ± 0.15 ^b^	12.10 ± 0.56 ^a,b^
	GLA + IR	2.32 ± 0.26 ^b,c^	0.67 ± 0.26 ^b^	16.48 ± 8.86	20.63 ± 0.53 ^b,c^	316.00 ± 76.15 ^b,c^	1.26 ± 0.24	9.50 ± 0.25 ^c^

Mice were pre-treated with 100 μg/kg GLA at 48, 24, and 1 h prior to irradiation, and were subjected to total-body irradiation of 7.5 Gy (at 1 Gy/min). Duodenum samples were obtained on day 1, day 3, day 7, and day 14 post-irradiation, and the level of antioxidants were estimated. Values (n = 6) expressed as (mean ± SE). Statistical significance analysis was performed by *t*-test (*p* < 0.001). Significance, expressed as ^a^, ^b^, ^c^, with control, GLA, and IR of same time point, respectively. C, control; GLA, gamma-linolenic acid; IR, irradiation (2 Gy).

## Data Availability

All data pertaining to this study are given in the manuscript.

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
