# Peer review of "Gamma-Linolenic Acid (GLA) Protects against Ionizing Radiation-Induced Damage: An In Vitro and In Vivo Study"

_biomolecules, 2022, doi:10.3390/biom12060797_

Round 1

Reviewer 1 Report

The lead author has published extensively on the biological effects of lipids. In this study, g-linolenic acid was demonstrated to have a protecting effect against ionizing radiation. Potential mechanisms of action were investigated in vitro and in vivo.

Some refinements will improve the manuscript-

1) The cell toxicity of PUFAs is well known. In reporting the toxicity in RAW 264.7, the authors should compare to data in other cell types and provide references.

2) Compound dosing should be performed at equimolar concentrations. Using g/mL quantities is imprecise - in this case the MW of the PUFAs ranges from 278 to 328.

3) As a control, it would be nice to see the effects from a saturated fatty acid.  Surprisingly, the authors do not cite their own related study (Arch Med Sci, 2020) on ionising radiation or discuss the new data in context.

4) g-Linoleic acid significantly increases cell viability in both the control (Fig 3) and after irradiation at 48 hours. The authors should comment on the extent of radiation protection as the effect already occurs in the control.

5) Statistical significance must be taken into account. I doubt the data are sufficiently precise to need reporting to two decimal places.

Author Response

Radiation and GLA

Reviewer # 1:

Some refinements will improve the manuscript-

  • The cell toxicity of PUFAs is well known. In reporting the toxicity in RAW 264.7, the authors should compare to data in other cell types and provide references.

Response; As suggested by the reviewer, this has been done. Please see line 720 and accompanying new references reference 55).

  • Compound dosing should be performed at equimolar concentrations. Using g/mL quantities is imprecise - in this case the MW of the PUFAs ranges from 278 to 328.

Response: This suggestion by the reviewer is valid. When we buy the PUFAs and their metabolites form the supplier, they are sent in terms of micrograms and mgs. Hence all subsequent dilutions are in micrograms and mgs only. So instead of converting these values into M, we expressed all used concentrations of PUFAs in micrograms or mgs only. To facilitate this conversion of micrograms and mgs to M, the M Wt of the fatty acids used is given at the end of the text.    

  • As a control, it would be nice to see the effects from a saturated fatty acid.  Surprisingly, the authors do not cite their own related study (Arch Med Sci, 2020) on ionising radiation or discuss the new data in context.

Response: As suggested by the reviewer, our previous study wherein we showed that saturated and monounsaturated fatty acid are ineffective in protecting cells from the toxic action of chemicals is discussed -see line 725 and reference 55. Our previous study published in AMS has also been quoted as suggested by the reviewer- see reference 47.

  • g-Linoleic acid significantly increases cell viability in both the control (Fig 3) and after irradiation at 48 hours. The authors should comment on the extent of radiation protection as the effect already occurs in the control.

Response: As suggested by the reviewer this aspect is discussed in the revised version. Please see line 742 and reference 60.

  • Statistical significance must be taken into account. I doubt the data are sufficiently precise to need reporting to two decimal places.

Response: The statistical significance was given as we obtained in our calculations.

Reviewer 2 Report

In this study, the authors analyzed the role of bioactive lipids especially the unsaturated fatty acid GLA and their metabolites, both pro- and anti-inflammatory products in the biological actions of radiation and radiation-induced injury and its resolution.

 It was shown that GLA reduced DNA damage and enhanced metabolic viability that led to an increase in the number of surviving cells in vitro after radiation. These in vitro beneficial actions were confirmed by in vivo studies that revealed that the survival of female C57BL/6J mice exposed to lethal radiation (survival ~ 20%) is significantly enhanced (to ~80%) by GLA treatment.

The analysis of different parameter for the in vitro and in vivo studies suggest that GLA protects cells/tissues from lethal dose of radiation by producing appropriate changes in the inflammation and its resolution in a timely fashion.

This work was processed systematically. The data for the optimal condition like doses and duration of irradiation, choice of fatty acids and concentration were determined at the beginning. Interesting in vivo and in vitro results have been demonstrated and they have been well analyzed and discussed.

I accept this manuscript after very small corrections

 -          Figure 10 presents the most important results. It is small and must be made larger with better quality.

 -          In figure 10, the n=12 or n=6 do not match the number of points in the Kaplan-Meier curve. What do the many points in the curve represent?

 -          Figure 14 after line 597 there are two rows with the days instead of one

 -          All abbreviations must have the same spelling in the manuscript. As an example LXA4 which was often written as LX A4. Even several spaces are too much (line 685, 719). Please check everything again.

Author Response

Radiation and GLA

Reviewer # 2

In this study, the authors analyzed the role of bioactive lipids especially the unsaturated fatty acid GLA and their metabolites, both pro- and anti-inflammatory products in the biological actions of radiation and radiation-induced injury and its resolution.

 It was shown that GLA reduced DNA damage and enhanced metabolic viability that led to an increase in the number of surviving cells in vitro after radiation. These in vitro beneficial actions were confirmed by in vivo studies that revealed that the survival of female C57BL/6J mice exposed to lethal radiation (survival ~ 20%) is significantly enhanced (to ~80%) by GLA treatment.

The analysis of different parameter for the in vitro and in vivo studies suggest that GLA protects cells/tissues from lethal dose of radiation by producing appropriate changes in the inflammation and its resolution in a timely fashion.

This work was processed systematically. The data for the optimal condition like doses and duration of irradiation, choice of fatty acids and concentration were determined at the beginning. Interesting in vivo and in vitro results have been demonstrated and they have been well analyzed and discussed.

I accept this manuscript after very small corrections

 -          Figure 10 presents the most important results. It is small and must be made larger with better quality.

Response: s suggested by the reviewer, Figure 10 has been expanded and made more clear.

 -          In figure 10, the n=12 or n=6 do not match the number of points in the Kaplan-Meier curve. What do the many points in the curve represent?

Response: The figure has been made more clear and bigger for clarity.

 -          Figure 14 after line 597 there are two rows with the days instead of one

Response: This has been modified.

 -          All abbreviations must have the same spelling in the manuscript. As an example LXA4 which was often written as LX A4. Even several spaces are too much (line 685, 719). Please check everything again.

Response: All abbreviation has been expanded as suggested at the first mention. LXA4 has been corrected.